# Inductive Alignment for Table Representation with Fidelity and Consistency

## Abstract

Representing symbolic, schema-diverse tables remains a fundamental challenge. Symbolic attributes often carry rich domain semantics, yet header formats and lexical expressions vary widely across tables, making it difficult for existing methods to maintain stable semantics. While language models capture semantic regularities, their token-level contextualization and sequential biases are misaligned with the bi-directional structure of tables, leading to sensitivity to schema and limited generalization across in-domain tables. Existing methods that seek to mitigate these limitations tend to favor either *fidelity*—preserving discriminative schema–value relationships—or *consistency*—maintaining robustness to lexical and structural variations—yet rarely achieving both. We argue that effective table representations must determine what should differ and what should remain the same across in-domain tables. To operationalize this principle, we introduce the header–value segment as a minimal, semantically coherent unit that captures both a header's functional role and the domain semantics of its value. Figure 1 illustrates how segment-level modeling aligns domain-coherent schema variants while separating entity-specific content. Building on this idea, we propose NAVI—Entropy-aware Alignment with Header–Value Induction—a segment-centric framework that balances fidelity and consistency. Across real-world in-domain tables, NAVI significantly outperforms existing baselines on discriminative and generative tasks, while producing stable and interpretable segment embeddings. The source code of NAVI is available at: https://anonymous.4open.science/r/navi.

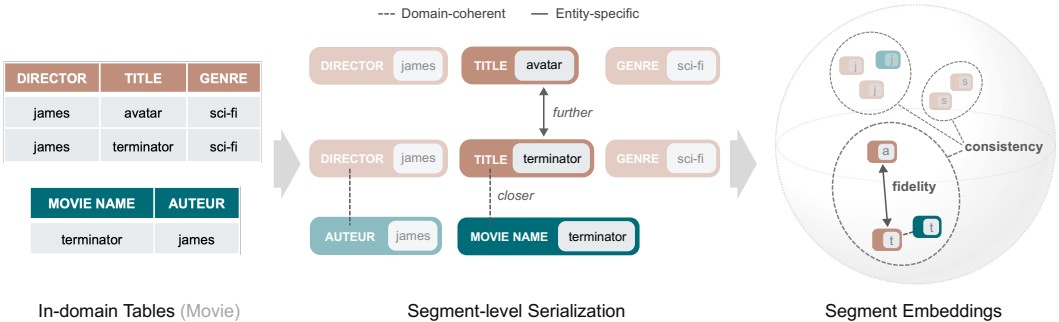

Figure 1: Illustration of **fidelity** and **consistency** in segment-level table representation. Left: In-domain movie tables exhibit heterogeneous schemas (e.g., DIRECTOR vs. AUTEUR) and symbolic attributes with shared domain semantics (e.g., GENRE). Middle: NAVI serializes each header–value pair into a *segment* and anchors headers using a global, context-free header encoder. Segments whose headers share similar semantics (e.g., DIRECTOR/AUTEUR or TITLE variants) are grouped accordingly, while segments containing different values under similar context (e.g., avatar vs. terminator) are also distinctly mapped. Right: After segment embedding, domain-coherent segments are pulled *closer* together, whereas entity-specific segments are pushed *further apart*. The figure visualizes how NAVI simultaneously enforces *consistency* (grouping lexical/structural variants) and *fidelity* (preserving discriminative entity-level semantics).

## 1 INTRODUCTION

**Motivation.** Tabular data encode information in a form fundamentally different from natural language. Headers instantiate domain-level semantics shared across rows and often across tables, yet they appear with substantial lexical noise, structural variation, and inconsistent formatting. While numerical attributes can be modeled reliably through type-aware techniques, symbolic attributes—categories, identifiers, and free-text values with widely varying cardinalities—carry much of a table's semantic content and require richer representational modeling.

Language models offer strong priors for symbolic data, but their sequential inductive biases make them ill-suited to the bi-directional layout of tables. They struggle with permutation invariance, heterogeneous schema patterns, and stable relational semantics across tables, leading to unstable header representations, sensitivity to superficial schema changes, and entanglement between schema-level concepts and row-specific content; a detailed discussion follows in the next subsection.

Classical pipelines such as gradient boosting decision trees Chen (2016) and numeric-specialized LM variants like TP-BERTa Yan et al. excel at capturing quantitative patterns, but they provide limited leverage for modeling symbolic schema–value semantics or generalizing across heterogeneous table structures. Consequently, neither traditional tabular models nor existing LM-based encoders adequately address the representational needs of symbolic, schema-diverse in-domain tables.

Overcoming these limitations requires preserving the semantic strengths of language models while correcting their structural blind spots. At the core is a fundamental question for table representation learning: *what should differ, and what should remain the same across in-domain tables?* This distinction yields two complementary desiderata. **Fidelity** determines the aspects to which representations must remain sensitive—preserving functional roles (*structural fidelity*) and maintaining entity-level distinctions (*domain fidelity*). **Consistency** determines the invariances representations must maintain—robustness to schema perturbations (*structural consistency*) and stable domain semantics across heterogeneous tables (*domain consistency*).

**Existing Works.** A large body of research adapts Transformer architectures to tabular data (Fang et al., 2024; Badaro et al., 2023), extending the success of sequence models on unstructured text (Vaswani et al., 2017; Devlin et al., 2019). These models serialize tables into token sequences and incorporate structural inductive biases—such as row/column embeddings or hierarchical encodings—to mitigate sequential biases. However, as in the typical trade-off between sensitivity and robustness in representation learning, existing methods struggle to achieve both *fidelity* and *consistency* simultaneously for in-domain tables.

On the one hand, fidelity-oriented methods (Herzig et al., 2020; Yin et al., 2020; Iida et al., 2021; Deng et al., 2022; Wang et al., 2021) explicitly model rows, columns, or tree structures to capture fine-grained schema information of a table. By contextualizing tokens according to their functional roles in a table with vertical or horizontal attention, these methods achieve strong *fidelity*. However, they compromise the consistency of table representations; vulnerability to schema variations undermines *structural consistency*, while table-specific designs hinder generalization across in-domain tables, weakening *domain consistency*.

On the other hand, consistency-oriented methods (Jung & Yoon, 2025) enforce schema stability by interpolating context-free header embeddings with contextualized ones and regularizing their distance. Although this anchoring achieves *consistency*, it is applied only to header tokens and relies on token-level contextualization to absorb value information. As a result, the learned headers become largely value-independent, producing overly smoothed header semantics and a header–value misalignment that weakens both *structural and domain fidelity* under schema diversity.

These limitations of existing methods fundamentally arise from the token-level contextualization of tabular data, similar to unstructured text, where table-specific adaptations function only as a limited workaround. Merely token-level encoding fails to accurately learn header-value relationships, undermining fidelity, and to be fully aware of schema or lexical variation to preserve consistency.

**Main idea and Contributions.** To bridge this gap, we introduce the concept of a header-value *segment*, a minimal yet semantically meaningful unit of a table that integrates structural roles with domain semantics. By treating the segment as the fundamental building block of representation

learning, models can encode the essence of in-domain tables into a unified embedding that simultaneously balances fidelity and consistency. Grounded in this concept, we propose a novel tabular embedding framework **NAVI**; ENtropy-aware Alignment with Header-Value Induction. NAVI aims to capture the structural properties of tables through *schema-aware segment induction and modeling*. It also employs *entropy-driven alignment of segments* to selectively incorporate domain knowledge shared among in-domain tables.

In summary, we make the following contributions: (1) We identify the two key desiderata, fidelity and consistency, as a principled foundation for effective in-domain table representation learning. (2) We introduce the notion of a header-value segment as the fundamental building block of tables, and propose NAVI, a novel segment-centric embedding framework, with a theoretical analysis for the two desiderata. (3) We conduct extensive experiments on real-world in-domain tables, showing that NAVI outperforms existing baselines in both discriminative and generative downstream tasks. In addition, qualitative analyses further demonstrate the effectiveness of NAVI.

## 2 METHODOLOGY

We present a three-stage framework for segment-grounded representation learning from tabular data. Our methodology consists of: (1) Schema-aware Segment Induction, which defines the header–value segment as a structural unit and incorporates context-free header embeddings to ensure structural consistency; (2) Masked Segment Modeling, which extends the masked language modeling (MLM) objective with balanced masking of headers and values to enforce fine-grained schema–value dependencies, thereby achieving structural fidelity; and (3) Entropy-driven Segment Alignment, which leverages column entropy to distinguish domain-defining from entity-defining attributes, applying cross-column and cross-row alignment to ensure domain consistency and domain fidelity.

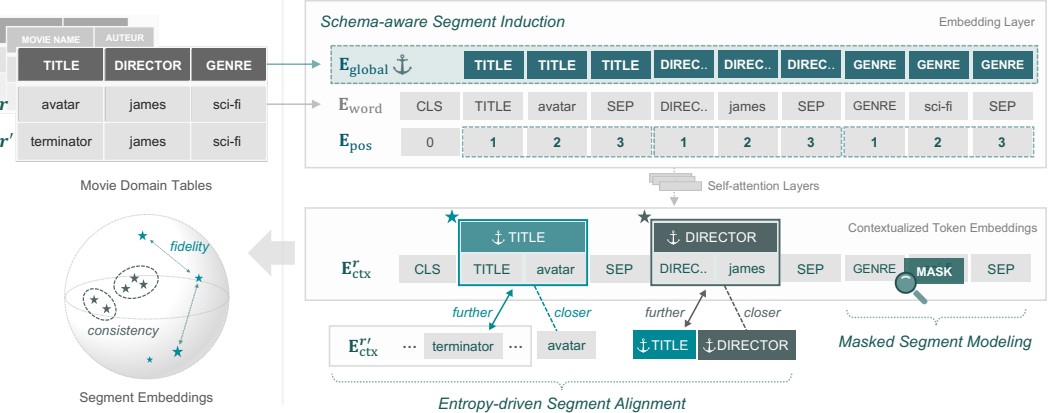

Figure 2: Overall procedure of NAVI. We optimize schema-induced representations for tokens and segments with masked modeling and entropy-driven alignment, preserving intra-table fidelity and inter-table consistency for in-domain tables.

### 2.1 SCHEMA-AWARE SEGMENT INDUCTION

**Header-Value Segment.** Unlike natural language, tables are inherently organized into rows, each corresponding to a distinct entity. To preserve this entity-level semantics while avoiding spurious cross-row interactions, we serialize each row independently as an unordered set of *header–value segments*. Each segment is constructed in the canonical form `header : value[SEP]` with a special `[CLS]` token prefixed at the row level. This segmentization provides an explicit structural unit that grounds schema semantics without imposing any ordering among columns.

For each segment, we identify header-token indices and value-token indices based on the tokenized sequence, excluding delimiter and special tokens. For example:

In the resulting token sequence, header tokens form the header span $\mathcal{P}_{\text{MOVIENAME}} = [1, 2]$, while value tokens form the value span $\mathcal{P}_{\text{harrypotter}} = [4, 5]$. The delimiter ":" is treated as a boundary marker and is omitted from both spans, and special tokens are likewise excluded. To prevent the model from encoding column order, each segment $k$ of length $m_k$ is assigned an independent positional reinitialization:

$$E_{\text{pos}}(x_j^{(k)}) = P_j, \quad j = 0, \ldots, m_k - 1,$$

where $P_j$ is a sinusoidal positional embedding. This yields both row and column permutation invariance, and together these invariances constitute the form of *structural consistency*.

**Global Header Representation.** To anchor header semantics consistently across contexts (e.g., rows, tables), we introduce a lightweight encoder dedicated for encoding header strings. Given a header tokenized as $h = [t_1, \ldots, t_n]$, the encoder produces self-attended embeddings $\{e_{t_1}, \ldots, e_{t_n}\}$, $e_{t_k} \in \mathbb{R}^d$. A single universal embedding for header $h$, $E_{\text{global}}(h) \in \mathbb{R}^d$ is then obtained by pooling, independent of any specific table context.

Unlike prior approaches that construct column embeddings by coupling headers with local values (Yin et al., 2020; Iida et al., 2021), our header representations remain context-free. This provides a consistent semantic anchor across diverse tables and serves as a supportive bias for domain consistency. It is further complemented by stronger distribution-level regularization through entropy-driven segment alignment.

**Header-conditioned Token Representation.** To provide a supportive bias toward domain-level consistency, we condition token embeddings on their corresponding global header representations. Specifically, $E_{\text{global}}(h)$ is added as a bias to each token $x_j^{(k)}$ within its segment:

$$z_j^{(k)} = E_{\text{word}}(x_j^{(k)}) + E_{\text{pos}}(x_j^{(k)}) + E_{\text{global}}(h_k).$$

These conditioned embeddings are contextualized by a transformer encoder, yielding token representations $\mathbf{e}_t \in \mathbb{R}^d$. This mechanism enforces a stable schema bias, promoting *structural fidelity* by maintaining schema-value dependencies under noisy or mutated tables.

Building on this token representation, we obtain header- and value-level contextual embeddings by mean-pooling the encoder outputs at their respective spans. The contextualized header embedding $H_{\text{ctx}}(r, h)$ is obtained by pooling over positions in $\mathcal{P}_h$, and the contextualized value embedding $V_{\text{ctx}}(r, h)$ is analogously obtained from positions in $\mathcal{P}_v$. These span-based contextual embeddings preserve the intended structural separation between schema and content while capturing their interactions within each row.

**Header-conditioned Segment Representation.** At the row level, we construct segment embeddings that integrate both schema anchors and contextualized representations. For header $h$ in row $r$, we first obtain contextualized token embeddings from the transformer encoder output. The contextualized header and value embeddings, respectively denoted as $H_{\text{ctx}}(r, h)$ and $V_{\text{ctx}}(r, h)$, are extracted by pooling the contextualized token embeddings at the positions of $h$ and its corresponding value.

Finally, the segment embedding concatenates the global header with row-aware components:

$$E_{\text{seg}}(r, h) = g\big(E_{\text{global}}(h) \,\|\, H_{\text{ctx}}(r, h) \,\|\, V_{\text{ctx}}(r, h)\big),$$

where $\|$ denotes concatenation, and $g(\cdot)$ is a projection network. Such integrated embeddings capture both schema–value dependencies and global semantics, thereby providing a relationally expressive foundation that promotes *domain fidelity* across heterogeneous tables.

## 2.2 MASKED SEGMENT MODELING

**Structure-based Masking.** Standard masked language modeling (MLM) has proven effective for natural language (Devlin et al., 2019), but its direct application to tables is suboptimal. Headers

and values are semantically different, and their dependencies are crucial for relational reasoning. By treating all tokens uniformly, Vanilla MLM risks overlooking the structural distinction between schema and content, undermining its ability to maintain *structural fidelity*. To address this, we introduce a structure-aware masked segment modeling (MSM) that explicitly models schema–value dependencies by partitioning each row of segments into three masking regimes:

- **Header-masked segments:** header tokens in selected segments are masked, forming the set $\mathcal{M}_h$. The model must recover header names from associated values.

- **Value-masked segments:** value tokens in selected segments are masked, forming the set $\mathcal{M}_v$. The model must infer values from headers and row context.

- **Vanilla MLM:** a random subset of remaining tokens is masked, forming $\mathcal{M}_r$. This acts as a regularization term that prevents overfitting to header–value co-occurrence patterns.

**Objective.** For each masked token $t \in \mathcal{M}$ with contextualized token embedding $\mathbf{e}_t$, the classifier produces a logit vector $\mathbf{z}_t = W\mathbf{e}_t + b \in \mathbb{R}^{|\mathcal{V}|}$. The masked segment modeling (MSM) loss is then given by the standard softmax cross-entropy:

$$\mathcal{L}_{\text{msm}} = -\frac{1}{|\mathcal{M}|} \sum_{t \in \mathcal{M}} \log \frac{\exp(\mathbf{z}_t[t])}{\sum_{v \in \mathcal{V}} \exp(\mathbf{z}_t[v])}, \quad \mathcal{M} = \mathcal{M}_h \cup \mathcal{M}_v \cup \mathcal{M}_r,$$

where $\mathbf{z}_t[v]$ is the logit corresponding to vocabulary item $v$. The MSM objective with structured masking compels the encoder to learn functional roles of tokens and schema–value dependencies, thereby realizing the *structural fidelity*.

## 2.3 ENTROPY-DRIVEN SEGMENT ALIGNMENT

**Entropy-based Column Categorization.** While the preceding methods ensure structural consistency and structural fidelity, they do not by themselves guarantee domain consistency or domain fidelity. To achieve these desiderata, we require an additional mechanism that explicitly aligns representations. Contrastive learning (Oord et al., 2018; Chen et al., 2020; Lee et al., 2022) has been widely used to arrange embeddings according to a target semantic objective, but the straightforward adoption—applying contrastive loss directly at the row (i.e., instance, entity) level—fails to distinguish between schema-level semantics and instance-specific attributes. This results in entangled representations that blur domain boundaries or collapse row-level distinctions.

To overcome this limitation, we propose an entropy-based column categorization. Instead of aligning rows indiscriminately, we categorize columns by the entropy of their empirical value distributions and use this categorization as the foundation for aligning segments and headers-values:

- **Domain-coherent columns** $\mathcal{H}_{\mathbf{dom}}$**:** low-entropy columns (e.g., below the 10 percentile) with stable domain-level concepts (e.g., genre and director in movie tables). Aligning their segments and headers enforces consistent semantics across tables, promoting *domain consistency*.

- **Entity-discriminative columns** $\mathcal{H}_{\mathbf{ent}}$**:** high-entropy columns (e.g., above the 90 percentile) with instance-specific attributes (e.g., title and url in movie tables). Aligning their segments and values enhances row separability within a domain, enhancing *domain fidelity*.

**Objective.** Given a query $q$, a positive sample $x^+$, a set of negative samples $\mathcal{X}^-$, and a temperature $\tau$, the InfoNCE objective (Oord et al., 2018) is set as:

$$\mathcal{L}_{\text{InfoNCE}}(q, x^+, \mathcal{X}^-, \tau) = -\log \frac{\exp(q \cdot x^+/\tau)}{\exp(q \cdot x^+/\tau) + \sum_{x^- \in \mathcal{X}^-} \exp(q \cdot x^-/\tau)}.$$

For headers in domain-coherent columns $h_{\text{dom}} \in \mathcal{H}_{\text{dom}}$, cross-header alignment matches segments with their global header embeddings. This ensures that headers representing similar domain concepts are consistently aligned across rows and tables. We optimize the domain-coherent loss:

$$\mathcal{L}_{\text{dom}}^t = \mathbb{E}_{r \sim \mathcal{R},\, h \sim \mathcal{H}_{\text{dom}}} \left[ \mathcal{L}_{\text{InfoNCE}}(q_{\text{dom}}(r, h), x_{\text{dom}}^+(h), \mathcal{X}_{\text{dom}}^-(h), \tau_{\text{dom}}) \right], \text{where}$$

$$q_{\text{dom}}(r, h) = E_{\text{seg}}(r, h),\ x_{\text{dom}}^+(h) = E_{\text{global}}(h),\ \mathcal{X}_{\text{dom}}^-(h') = \{ E_{\text{global}}(h') \mid h' \in \mathcal{H}_{\text{dom}},\ h' \neq h \}.$$

For headers in entity-discriminative columns $h_{\text{ent}} \in \mathcal{H}_{\text{ent}}$, cross-row alignment matches segments with row-aware values. This encourages row-level separability by ensuring distinct rows in the same table remain distinguishable. We optimize the entity-discriminative loss:

$$\mathcal{L}_{\text{ent}}^t = \mathbb{E}_{r\sim\mathcal{R},\ h\sim\mathcal{H}_{\text{ent}}}\big[\mathcal{L}_{\text{InfoNCE}}(q_{\text{ent}}(r,h), x_{\text{ent}}^+(h), \mathcal{X}_{\text{ent}}^-(h), \tau_{\text{ent}})\big], \text{where}$$

$$q_{\text{ent}}(r,h) = E_{\text{seg}}(r,h),\ x_{\text{ent}}^+(r,h) = V_{\text{ctx}}(r,h),\ \mathcal{X}_{\text{ent}}^-(r,h) = \{\, V_{\text{ctx}}(r',h) \mid r' \in \mathcal{R},\ r' \neq r \,\}.$$

Finally, given a batch $\mathcal{B}$ in input tables and a balancing parameter $\lambda_{\text{align}}$ for generative and discriminative supervision, the overall training objective is formulated as:

$$\mathcal{L}_{\text{total}} = \mathcal{L}_{\text{msm}} + \lambda_{\text{align}} \cdot \mathcal{L}_{\text{align}}, \text{where } \mathcal{L}_{\text{align}} = 1/|\mathcal{B}| \cdot \sum_{t\in\mathcal{B}}(\mathcal{L}_{\text{dom}}^t + \mathcal{L}_{\text{ent}}^t).$$

Appendix A discusses the theoretical analysis of the schema induction and contrastive alignment.

## 3 Experiments

### 3.1 Experimental Setup

**Datasets.** We evaluate on four datasets from two domains, Movie and Product. For pretraining, we use subsets of WDC WebTables (Peeters et al., 2024), selecting the 100 largest tables per domain—WDC Movie (480,817 rows) and WDC Product (3,930,877 rows)—and subsample 480,817 rows from each for balance. To ensure compatibility with BERT-style models, all tables are processed through a standardized pipeline (see Appendix D.2). For downstream evaluation, we construct held-out subsets of 45,000 rows per domain ($\approx$10% of pretraining). We uniformly subsample 1,000 rows per each evaluation run for consistency and efficiency.

**Baseline Methods.** We evaluate our approach against representative table embedding models spanning major paradigms. BERT serves as the generic transformer backbone underlying most language model–based table encoders; its performance highlights the limitations of applying vanilla language models to tabular data. TAPAS, the most widely adopted table encoder, exemplifies fidelity-oriented approaches, while HAETAE represents a consistency-oriented encoder. This selection enables a systematic comparison of their strengths and limitations with respect to fidelity and consistency.

**Implementaion.** We configure NAVI to balance domain and structural objectives. For domain objectives, we set contrastive temperature $\tau$ to 0.02 for entity-discriminative columns and 0.14 for domain-coherent columns, and vary the alignment weight $\lambda_{\text{align}}$ across tasks. For structural objectives, we adjust the header–value–baseline (H:V:B) masking ratio. We use $\lambda_{\text{align}}$ as 0.05 and H:V:B = 4:4:2 as the default. Further details and sensitivity analysis appear in Appendix D. All models are trained on the same datasets for 2 epochs with a batch size of 32, AdamW (Loshchilov & Hutter) with a learning rate of $3 \times 10^{-5}$, and a weight decay of 0.01.

### 3.2 Fidelity Analysis

We evaluate *fidelity*, the faithfulness of representations to table semantics. Fidelity spans two dimensions: *domain fidelity*, which preserves entity-level discriminability, and *structural fidelity*, which models schema–value dependencies within rows. We probe domain fidelity through **discriminative tasks** (Row Classification and Row Clustering) and structural fidelity through **generative tasks** (Value Imputation and Header Prediction).

**Discriminative Tasks.** To assess *domain fidelity*—whether embeddings preserve entity-level separability—we evaluate *Row Classification* and *Row Clustering*. For LM-based table encoders (BERT, TAPAS, HAETAE, and NAVI), each row is serialized and encoded once; the final `[CLS]` embedding is used as a feature vector for downstream evaluation. Classification uses 10 balanced classes per domain (top product categories, top movie genres; $\sim$1,000 samples), reporting Macro-F1 from XGBoost, Logistic Regression, and TabPFN. Clustering probes the same label space via Agglomerative, scored by Silhouette and B³-F1, with all results averaged over 5 subsampled runs. As shown in Table 1, BERT relies on shallow cues and achieves modest discriminability, while

Table 1: Performance on discriminative tasks. The table shows results from [CLS] token embeddings. Macro-F1 scores for classification (using XGBoost, Logistic Regression, TabPFN) and Silhouette and B³-F1 scores for clustering (using Agglomerative).

| | Product | | | | Movie | | | |
| | R-Cls (F1) | | | R-Clt (Sil/ B³) | R-Cls (F1) | | | R-Clt (Sil/ B³) |
| Model | XGB | LR | PFN | Agglo. | XGB | LR | PFN | Agglo. |
|---|---|---|---|---|---|---|---|---|
| BERT | 0.915 | 0.931 | 0.938 | 0.215 / 0.605 | 0.597 | 0.653 | 0.647 | 0.076 / 0.274 |
| TAPAS | 0.927 | 0.934 | 0.932 | 0.406 / 0.770 | 0.607 | 0.665 | **0.675** | 0.087 / **0.320** |
| HAETAE | 0.916 | **0.942** | 0.938 | 0.233 / 0.663 | 0.607 | 0.634 | 0.663 | 0.073 / 0.279 |
| NAVI | **0.930** | 0.941 | **0.945** | **0.424** / **0.833** | **0.639** | **0.667** | 0.670 | **0.100** / 0.297 |
| Raw | 0.888 | 0.780 | 0.882 | – / – | 0.464 | 0.415 | 0.498 | – / – |
| TableVectorizer | 0.933 | 0.909 | 0.940 | – / – | 0.618 | 0.540 | N/A | – / – |
| NAVI$_{text\_emb+num}$ | **0.942** | 0.941 | 0.943 | – / – | **0.641** | 0.658 | 0.669 | – / – |

HAETAE's rigid anchoring suppresses value-sensitive variation. TAPAS improves fidelity but remains schema-sensitive. By contrast, NAVI consistently leads across classifiers and clustering, yielding compact and coherent row manifolds—demonstrating that entropy-driven alignment mitigates row collapse and strengthens entity-level fidelity under schema diversity.

Beyond LM-based encoders, we also evaluate non–LM baselines on the classification task—namely raw-feature classifiers (XGBoost, Logistic Regression, TabPFN) and TableVectorizer. TableVectorizer is a type-aware feature–engineering pipeline that scales numeric fields, encodes temporal attributes, applies one-hot encoding to low-cardinality text, and uses SentenceTransformer embeddings for high-cardinality text. Although powerful, this design produces very high-dimensional vectors in text-heavy domains such as Movie (often exceeding 2,000 dimensions), making it incompatible with TabPFN (N/A) and substantially larger than NAVI's 768-dimensional embeddings—yet without surpassing NAVI. This suggests that NAVI's performance could further improve with more complex encoder backbones Warner et al. (2025).

At the same time, TableVectorizer's strong performance demonstrates the value of type-aware feature engineering, particularly its explicit handling of numerical and temporal fields. Its main limitation lies in relying on a generic natural-language encoder (SentenceTransformer) for high-cardinality text. Replacing this component with a table-specialized LM encoder is therefore a promising direction. We illustrate this with a simple hybrid prototype, NAVI$_{text\_emb+num}$, which concatenates raw numerical features with NAVI's text-derived segments and already surpasses both raw-feature and the TableVectorizer pipeline. This indicates that NAVI can serve as a drop-in replacement for generic text encoders in feature-engineering pipelines, potentially yielding even stronger performance.

**Generative Tasks.** We examine *structural fidelity*, i.e., whether embeddings capture schema–value dependencies, through two tasks: *Header Prediction* and *Value Imputation*, respectively recovering masked headers and values from contextualized row tokens. We compare NAVI against BERT and HAETAE, which are naturally suited for generative tasks, but exclude TAPAS as its QA-oriented pretraining objective makes it infeasible for this setting. As shown in Table 2, NAVI achieves near-perfect header prediction, validating its global header encoder as a stable semantic anchor, and also outperforms in value imputation, where header-conditioned representations and structure-aware masking reinforce schema–value dependencies.

Table 2: Generative tasks.

| Model | Product | Movie |
|---|---|---|
| *Header* | | |
| BERT | 0.9284 | 0.9159 |
| HAETAE | 0.9219 | 0.9120 |
| NAVI | **0.9995** | **0.9990** |
| *Value* | | |
| BERT | 0.7586 | 0.6809 |
| HAETAE | 0.7735 | 0.6879 |
| NAVI | **0.7902** | **0.7077** |

## 3.3 CONSISTENCY ANALYSIS

We next evaluate *consistency*, the stability of representations under schema diversity. Consistency has two dimensions: *domain consistency* and *structural consistency*, which together denote invariance to lexical and structural diversity. Domain consistency is measured by clustering semantically

equivalent headers (e.g., `director` vs. `auteur`) using agglomerative clustering, with quality assessed by B³-F1 and NMI. Structural consistency is measured by permuting rows and computing the permutation sensitivity index (PSI $= \mathbb{E}_k[1 - \cos(z, \tilde{z}^{(k)})]$), where $z$ is the original row embedding and $\tilde{z}^{(k)}$ its $k$-th permutation, using both CLS and mean pooling. Consistent models should form compact header clusters and yield low PSI.

Table 3: Domain consistency of header clustering (H-Clt) is evaluated by B³-F1 and NMI (higher is better), and structural consistency of row permutation is evaluated with PSI (lower is better).

| Model | Product | | Movie | |
|---|---|---|---|---|
| | **H-Clt** (B³-F1 / NMI) | **PSI** (cls / mean) | **H-Clt** (B³-F1 / NMI) | **PSI** (cls / mean) |
| BERT | 0.7317 / 0.8749 | 6.35 e-2 / 7.59 e-3 | 0.6969 / 0.8798 | 5.81 e-2 / 6.61 e-3 |
| TAPAS | 0.7239 / 0.8750 | 1.24 e-2 / 6.70 e-3 | 0.6759 / 0.8726 | 1.01 e-2 / 6.32 e-3 |
| HAETAE | 0.7268 / 0.8742 | 6.15 e-2 / 8.70 e-3 | 0.7276 / 0.8864 | 6.53 e-2 / 7.83 e-3 |
| NAVI | **0.7920 / 0.9005** | **9.55 e-8 / 1.97 e-8** | **0.7996 / 0.9144** | **1.15 e-6 / 1.96 e-8** |

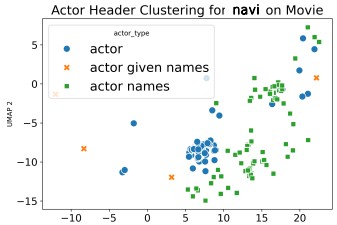

**Lexical Diversity.** On header clustering, NAVI yields the most coherent groups. HAETAE is competitive but still weaker than NAVI's alignment. Figure 3 illustrates this for the `actor` set: under NAVI (top), lexical variants converge into one cluster, while under BERT (bottom) they remain split. This contrast shows BERT encodes surface forms, whereas NAVI collapses aliases into canonical representations. Thus, the induction with contrastive alignment enforces domain consistency.

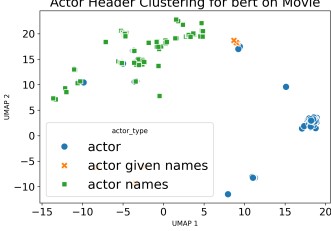

**Structural Diversity.** As shown in Table 3, BERT and HAETAE exhibit high PSI, indicating that their row representations remain sensitive to positional or ordering artifacts. TAPAS reduces this sensitivity but still preserves measurable permutation effects. In contrast, NAVI yields PSI values that are one to two orders of magnitude lower, approaching true invariance to row permutations. This shows that schema induction suppress spurious structural cues, producing stable row embeddings.

Figure 3: Header embeddings.

### 3.4 ABLATION STUDY

To disentangle the contribution of each component in NAVI, we organize our analysis along the four desiderata of our evaluation framework. On the fidelity side, we assess *domain fidelity* with Row Classification (R-Cls) and *structural fidelity* with Value Imputation (Val). On the consistency side, we measure *domain consistency* with Header Clustering (H-Clt) and *structural consistency* with the Permutation Sensitivity Index (PSI). This one-to-one mapping provides a clear lens into how Schema-aware Segment Induction (SSI), Structure-aware MSM (SMSM), and Entropy-driven Segment Alignment (ESA) each contribute to representations that are both faithful and consistent.

Table 4: Classification (Logistic Regression - F1), Accuracy for Value Imputation, Header Clustering (Agglomerative - NMI), Permutation Sensitivity Index (computed from cls row embeddings) across Product and Movie domains.

| Variant | Product | | | | Movie | | | |
|---|---|---|---|---|---|---|---|---|
| | **R-Cls** | **Val** | **H-Clt** | **PSI** | **R-Cls** | **Val** | **H-Clt** | **PSI** |
| **NAVI** | **0.9412** | **0.7902** | 0.9005 | **9.55e-8** | **0.6670** | 0.7077 | 0.9144 | **1.15e-8** |
| w/o SSI | 0.9137 | 0.2627 | 0.7071 | 1.35e-7 | 0.4899 | 0.2522 | 0.4374 | 1.52e-7 |
| w/o MSM | 0.9091 | 0.7659 | 0.8989 | 2.45e-7 | 0.5790 | 0.6927 | 0.9018 | 9.63e-8 |
| w/o ESA | 0.9358 | 0.7897 | **0.9007** | 1.92e-7 | 0.6488 | **0.7086** | **0.9149** | 9.26e-8 |

Results in Table 4 clarify how each module of NAVI sustains our four desiderata. Removing SSI yields the most severe degradation, collapsing Value Imputation and Header Clustering, which confirms schema anchoring as the linchpin of both fidelity and consistency. Removing MSM primarily affects structural fidelity and domain fidelity: value imputation and row classification drop, but consistency metrics (H-Clt, PSI) remain relatively stable, confirming that structure-aware masking mainly supervises schema–value functional roles rather than cross-table alignment. Dropping ESA leads to a clear loss in domain fidelity (R-Cls 0.9412→0.9358; 0.6670→0.6488), while the small gains in consistency are effectively negligible (H-Clt 0.9005→0.9007; 0.9144→0.9149). This shows that ESA plays a decisive role in enhancing row-level discrimination and preserving entity-specific distinctions. Overall, SSI provides the consistent structural backbone, MSM enforces structural fidelity, and ESA substantially strengthens domain fidelity.

## 3.5 QUALITATIVE ANALYSIS

Figure 4 visualizes segment embeddings from BERT and NAVI, both trained on the Movie domain. For BERT, segment embeddings are constructed by mean-pooling the contextualized token embeddings of each header–value span, ensuring comparability with NAVI's segment-level outputs. We project the resulting embeddings using t-SNE.

The geometry of BERT's segment space reflects the absence of explicit header–value alignment. Although high- and low-entropy segments appear almost linearly separable, their internal organization lacks meaningful semantic structure. Low-entropy segments—those expected to encode stable, domain-level concepts—scatter across the space with inconsistent shapes and densities, suggesting that BERT fails to form coherent semantic anchors. This instability is consistent with our hypothesis that contextual embeddings conflate schema-level semantics with row-specific fluctuations.

High-entropy segments, which should preserve fine-grained row identity, instead fragment into numerous compact, table-specific micro-clusters driven by superficial lexical or structural cues. Consequently, these segments become entangled with table-conditioned schematic patterns rather than maintaining consistent entity representations across tables. This behavior reflects the lack of domain fidelity: segments representing distinct rows are pulled together or separated in ways that mirror table artifacts rather than underlying semantics.

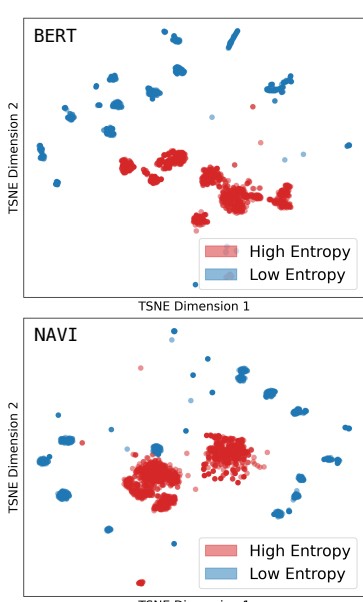

Figure 4: Visualization of segment embeddings from five heterogeneous Movie tables.

In contrast, NAVI induces a markedly different and more principled structure, shaped by its entropy-driven segment alignment objective. Low-entropy segments concentrate into tight, well-localized clusters that function as domain anchors. These clusters reflect NAVI's cross-header alignment for domain-coherent columns, where global header embeddings act as stable semantic centroids. The resulting contraction toward these centroids demonstrates that NAVI successfully extracts consistent schema-level semantics while filtering out table-specific noise.

High-entropy segments—aligned through row-level contrastive signals rather than schema-level induction—exhibit broader, more dispersed clusters. Instead of collapsing or fragmenting by table identity, they maintain coarse separation while remaining grounded in the shared domain space. This behavior arises because high-entropy alignment is anchored to each specific column: segments repel one another to preserve row-level distinctiveness, yet do not drift away from the semantic manifold defined by the domain. Importantly, these clusters do not collapse even when similar contextual patterns occur within the same table, illustrating NAVI's ability to preserve instance-level variability without overfitting to table-specific quirks.

## 4 CONCLUSION

In this paper, we revisit table representation through the two principled desiderata, fidelity and consistency, and exploit the header–value segment as the atomic unit to balance them. NAVI implements this idea with (i) Schema-aware Segment Induction (SSI) that builds segment embeddings anchored by a global, context-free header encoder, (ii) Masked Segment Modeling (MSM) that enforces fine-grained schema–value dependencies, and (iii) Entropy-driven Segment Alignment (ESA) that aligns domain-coherent columns while preserving separation for entity-discriminative ones. Empirical studies demonstrated that NAVI achieves higher performances on both header prediction and value imputation, in addition to consistent gains on classification and clustering tasks. Qualitatively, the resulting embedding space exhibits a core–periphery geometry (i.e., a shared semantic core for stable headers and a flexible periphery for instance-specific attributes) in accordance with our learning objectives. Ablation studies also confirm that the efficacy of the three main components: SSI as the building blocks for fidelity and consistency, MSM for schema–value coupling, and ESA for permutation stability and row discriminability. Together, these results position NAVI as a segment-centric, alignment-guided alternative to existing token-oriented encoders, narrowing the gap between symbolic tabular data and contextualized representations. We believe this work opens up practical opportunities and future work for applications with LLM-table interactions, such as question answering and retrieval-augmented generation on in-domain tables.

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

# A    THEORETICAL ANALYSIS

## A.1    SCHEMA INDUCTION: A MECHANISTIC ANALYSIS OF STRUCTURAL PROPERTIES

We analyze our **Schema-aware Segment Induction**, a theoretically grounded mechanism that introduces two inductive biases essential for table representation learning: (1) *Header–Value Coupling*, which enforces schema–value dependencies and preserves token roles, thereby realizing **Structural Fidelity**; and (2) *Segment-Order Equivariance*, which treats rows as sets of header–value segments and removes spurious order dependence, thereby realizing **Structural Consistency**.

### STRUCTURAL FIDELITY VIA SCHEMA-CONSISTENT ATTENTION ROUTING

**Setup.**    For segment $k$ with header $h^{(k)}$, let the token representation be

$$z_p = z_{\text{base}} + E_{\text{global}}(h^{(k)}),$$

where $z_{\text{base}}$ contains word and positional embeddings and $E_{\text{global}}(h^{(k)})$ is the universal header embedding. Queries and keys are linear maps $Q_p = W_Q z_p$, $K_q = W_K z_q$.

**Analysis.**    Let the token representation for $p$ in segment $k$ be

$$z_p = b_p + E, \qquad b_p := z_{\text{base},p}, \quad E := E_{\text{global}}(h^{(k)}).$$

With $Q_p = W_Q z_p$, $K_q = W_K z_q$ and $M := W_Q^\top W_K$, the attention logit expands to

$$\begin{aligned}
\ell_{pq} = Q_p^\top K_q &= (b_p + E)^\top M (b_q + E) \\
&= b_p^\top M b_q + b_p^\top M E + E^\top M b_q + E^\top M E.
\end{aligned} \tag{1}$$

The quadratic term $E^\top M E$ is independent of $(p, q)$ and thus acts as a *shared, header-dependent bias* within the entire segment $k$. The two cross-terms vary with $p, q$, but under LayerNorm (Ba et al., 2016) ($\mathbb{E}[b_p] = \mathbb{E}[b_q] = 0$) their expectations vanish. Hence the expected logit decomposes as

$$\mathbb{E}_{p,q}\, \ell_{pq} = \mathbb{E}_{p,q}[b_p^\top M b_q] + E^\top M E,$$

where the second term is the segment-wide bias.

Gradient w.r.t. $E$, Differentiating equation 1 gives

$$\nabla_E \ell_{pq} = M b_q + M^\top b_p + (M + M^\top) E.$$

Averaging over all $(p, q)$ within the segment yields

$$\mathbb{E}_{p,q}\, \nabla_E \ell_{pq} = (M + M^\top) E,$$

since $\mathbb{E}[b_p] = \mathbb{E}[b_q] = 0$. Thus, the expected update direction is the same for all tokens in the segment, depending only on $E$ and the projection matrices.

Since the MSM loss is token-level cross-entropy and analytically unwieldy, we study a surrogate quadratic objective $\mathcal{J}(E)$ that isolates the effect of header embeddings on attention logits.

$$\mathcal{J}(E) := \sum_{p,q} \ell_{pq} = \sum_{p,q} b_p^\top M b_q + 2 E^\top M \sum_q b_q + |S_k|^2 E^\top \text{Sym}(M) E,$$

where $|S_k|$ is the number of tokens in the segment. The stationary point satisfies

$$\nabla_E \mathcal{J}(E) = 2M \sum_q b_q + 2|S_k|^2 \text{Sym}(M) E = 0,$$

so that

$$E^\star = -\big(|S_k|^2 \, \text{Sym}(M)\big)^{-1} M \sum_q b_q.$$

With LayerNorm, $\sum_q b_q \approx 0$, making the optimizer align with the quadratic term $E^\top \text{Sym}(M) E$.

**Conclusion.**    Adding $E_{\text{global}}(h^{(k)})$ to all tokens yields a shared quadratic bias $E^\top \text{Sym}(M) E$ independent of values, and a uniform update direction $(M + M^\top) E$. Together, these reinforce schema–value coupling consistently across tokens in a segment, ensuring **Structural Fidelity**.

## STRUCTURAL CONSISTENCY VIA EQUIVARIANCE

**Setup.** Each row is serialized as a set of header–value segments $\{s(r, h_k)\}$, with segment-wise positional encodings but no global positions. Thus the encoder $g(\cdot)$ processes each segment independently, without reference to their global order.

**Analysis.** Since each segment is processed locally, the encoder $g$ is *permutation-equivariant*:

$$g(\pi \cdot \{s(r, h_k)\}) = \pi \cdot g(\{s(r, h_k)\}).$$

For any permutation $\pi$, the encoder output followed by a permutation-invariant readout $\rho$, specifically mean pooling over segment embeddings, satisfies

$$f_{\mathrm{mean-pool}}(r) = \rho\left(\sum_k \phi(s(r, h_k))\right),$$

which matches the functional form of Deep Sets (Zaheer et al., 2017), with $\phi = g$ and $\rho$ the pooling. By the universal approximation theorem for Deep Sets, $f_{\mathrm{mean}}$ can approximate any continuous permutation-invariant function over sets of segments.

Let $z_{\mathrm{cls}}$ be the row token. One self–attention update is

$$z'_{\mathrm{cls}} = \sum_q \alpha_{\mathrm{cls} \to q} V_q, \qquad \alpha_{\mathrm{cls} \to q} = \frac{\exp(\ell_{\mathrm{cls},q}/\tau)}{\sum_{q'} \exp(\ell_{\mathrm{cls},q'}/\tau)}, \qquad \ell_{\mathrm{cls},q} = (W_Q z_{\mathrm{cls}})^\top (W_K z_q). \quad (2)$$

For any permutation $\pi$ of segments in the row, the value/key sequences are merely reindexed:

$$\{(z_q, V_q)\}_q \mapsto \{(z_{\pi(q)}, V_{\pi(q)})\}_q \quad \Rightarrow \quad \{\ell_{\mathrm{cls},q}\}_q \mapsto \{\ell_{\mathrm{cls},\pi(q)}\}_q \quad \Rightarrow \quad \{\alpha_{\mathrm{cls} \to q}\}_q \mapsto \{\alpha_{\mathrm{cls} \to \pi(q)}\}_q.$$

Plugging the reindexed weights/values into equation 2 gives

$$z'_{\mathrm{cls}}(\pi \cdot \{s(r, h_k)\}) = \sum_q \alpha_{\mathrm{cls} \to \pi(q)} V_{\pi(q)} = \sum_q \alpha_{\mathrm{cls} \to q} V_q = z'_{\mathrm{cls}}(\{s(r, h_k)\}),$$

so the CLS update is permutation *invariant* when the operation is a pure reindexing (no extra biases, identical residual paths, exact arithmetic).

*Relaxation to $\varepsilon$–stability.* In practice, residual connections, layernorm/biases and finite precision introduce small deviations. We measure these by the permutation sensitivity index (PSI):

$$\mathrm{PSI} = \mathbb{E}_\pi\big[1 - \cos\big(f(r), f_\pi(r)\big)\big],$$

with $f(r)$ the row embedding (CLS or mean-pooled) and $f_\pi(r)$ after permuting segments by $\pi$. We say the encoder is *$\varepsilon$–permutation-stable* if $\mathrm{PSI} \le \varepsilon$.

**Conclusion.** Mean pooling yields $f(r) = \rho(\sum_k \phi(s(r, h_k)))$, targeting invariance. For CLS, the derivation above shows invariance in the ideal limit and *strong approximate* invariance in practice, with $\varepsilon$ empirically small (Table 3). Hence both readouts achieve **Structural Consistency**.

## A.2 CONTRASTIVE ALIGNMENT: GEOMETRIC FOUNDATIONS OF DOMAIN PROPERTIES

We analyze Entropy-driven Segment Alignment, an InfoNCE-based objective that provably induces domain manifold geometry in the segment embedding space. Building on the alignment–uniformity framework of Wang & Isola (2020), we show:

- Low-entropy (domain-coherent) columns are contracted toward its corresponding semantic centroids, forming a cross-table domain anchors rather than table-specific clusters. This realizes entropy-aware alignment and thereby **Domain Consistency**.
- High-entropy (entity-specific) columns experience entropy-aware uniformity: ESA repels rows away from one another, but in a controlled manner that keeps them contained within the same domain. This realizes **Domain Fidelity** by preserving row-level separability.

These guarantees provide the theoretical foundation for the empirical patterns in Figure 4, low entropy (schema-stable) segments contract onto semantic centroids as anchors, forming a cross-table domain manifold, while high-entropy (entity-specific) segments spread across this manifold in a uniform manner.

### PRELIMINARIES

Let $(\mathcal{T}, \mathcal{F}, P)$ be a probability space over tables, where $\mathcal{T}$ denotes the set of admissible tables, $\mathcal{F}$ is a $\sigma$-algebra, and $P$ is a probability measure capturing the empirical distribution of tables. An encoder $f_\theta : \mathcal{T} \to \mathcal{V}$ maps each table $T \in \mathcal{T}$ into a metric space $(\mathbb{R}^d, D)$, where $\mathcal{V}$ denotes the representation space endowed with distance $D$. We adopt the following assumptions:

(A1) (**Normalization**) All embeddings $E_{\text{seg}}(\cdot)$, $V_{\text{ctx}}(\cdot)$, and $E_{\text{global}}(\cdot)$ are $\ell_2$-normalized, i.e., lie on the unit sphere $\mathbb{S}^{d-1} \subset \mathbb{R}^d$.

(A2) (**Geometry**) The distance metric is the cosine distance $D(u, v) = 1 - u^\top v$, inducing a geodesic structure consistent with the sphere.

(A3) (**Information-Theoretic Objective**) The InfoNCE loss uses in-batch negative sampling sufficiently dense over rows, approximating a variational lower bound on mutual information.

(A4) (**Optimization**) The temperature $\tau > 0$ is fixed, scaling contrastive forces smoothly.

(A5) (**Entropy Estimation**) Column entropy is estimated from the empirical distribution. Misclassification probability decays exponentially in the number of rows (via large deviation bounds).

### ENTROPY-AWARE ALIGNMENT AND UNIFORMITY

Following Wang & Isola (2020), contrastive learning can be understood via two functionals: *alignment*, the expected closeness of positive pairs, and *uniformity*, the spreading of representations across the unit sphere. We adapt these notions by conditioning on column entropy.

**Notation.** For a column $c$, let $\mu_c := E_{\text{global}}(h_c)/\|E_{\text{global}}(h_c)\|$ be its normalized global header (semantic centroid). Let $\mathcal{N}_{\text{cent}}(c) = \{\mu_{c'} : c' \neq c\}$ denote centroids of other headers. For high-entropy columns, let $v(r, h_c) := V_{\text{ctx}}(r, h_c)/\|V_{\text{ctx}}(r, h_c)\|$ be the normalized contextual value.

**Definition 1** (Domain Consistency (entropy-aware alignment)). *For $c \in \mathcal{C}_{\text{low}}$, positives are centroid pairs $(s(r, h_c), \mu_c)$. Define*

$$\mathcal{L}_{\text{align}}^{\text{low}}(f; \alpha) := \mathbb{E}_r \|s(r, h_c) - \mu_c\|_2^\alpha.$$

*The model is $\epsilon_{\text{con}}$-consistent if $\mathcal{L}_{\text{align}}^{\text{low}} \leq \epsilon_{\text{con}}$.*

**Definition 2** (Domain Fidelity (entropy-aware uniformity)). *For $c \in \mathcal{C}_{\text{high}}$, the positive is $(s(r, h_c), v(r, h_c))$ and negatives are $(s(r, h_c), v(r', h_c))$ with $r' \neq r$. Dispersion is measured by*

$$\mathcal{L}_{\text{unif}}^{\text{high}}(f; t) := \log \mathbb{E}_{r \neq r'} \exp\big(-t\|s(r, h_c) - s(r', h_c)\|_2^2\big).$$

*The model is $\epsilon_{\text{dom}}$-faithful if $\mathcal{L}_{\text{unif}}^{\text{high}} \geq -\epsilon_{\text{dom}}$.*

**Assumptions for Entropy Partition.** $\mathcal{C}_{\text{low}} = \{c : H(c) \leq H_0\}$ and $\mathcal{C}_{\text{high}} = \{c : H(c) \geq H_1\}$ with $H_0 < H_1$. For $c \in \mathcal{C}_{\text{low}}$, positives are $(s(r, h_c), \mu_c)$ and negatives are $(s(r, h_c), \mu_c^-)$ with $\mu_c^- \in \mathcal{N}_{\text{cent}}(c)$. For $c \in \mathcal{C}_{\text{high}}$, positives are $(s(r, h_c), v(r, h_c))$ and negatives are $(s(r, h_c), v(r', h_c))$.

**Assumption 1 (MI gap – centroid/value forms).** There exist $\Delta_{\text{pos}}, \Delta_{\text{neg}} > 0$ s.t.

$$\mathbb{E}\langle s, \mu_c \rangle - \mathbb{E}\langle s, \mu_c^- \rangle \geq \Delta_{\text{pos}} \quad (c \in \mathcal{C}_{\text{low}})$$

$$\mathbb{E}\langle s,\ v(r', h_c) \rangle - \mathbb{E}\langle s,\ v(r, h_c) \rangle \geq \Delta_{\text{neg}} \quad (c \in \mathcal{C}_{\text{high}}).$$

**Assumption 2 (Entropy estimation).** $\Pr\big(\sup_c |\widehat{H}(c) - H(c)| \leq C\sqrt{\frac{\log(1/\delta)}{m_c}}\big) \geq 1 - \delta$.

**Theorem 1** (Domain Consistency–Fidelity Guarantee). *Suppose Assumptions 1–2 hold and let $\theta^\star$ satisfy $\mathcal{L}_{\text{align}}(\theta^\star) \leq \eta$. Then there exist functions $\phi_1, \phi_2$ with $\phi_i$ nondecreasing in $\eta, \tau$ and nonincreasing in $B$ such that*

$$\sup_{c \in \mathcal{C}_{\text{low}}} \mathcal{L}_{\text{align}}^{\text{low}}(f; \alpha) \leq \phi_1(\eta, \tau, B) = \frac{1}{\Delta_{\text{pos}}} \psi_1(\eta, \tau, B), \tag{3}$$

$$\inf_{c \in \mathcal{C}_{\text{high}}} \mathcal{L}_{\text{unif}}^{\text{high}}(f; t) \geq \phi_2(\eta, \tau, B) = \frac{1}{\Delta_{\text{neg}}} \psi_2(\eta, \tau, B), \tag{4}$$

*where $\psi_i(\eta, \tau, B) = (\tau(\eta - \log B))_+$. Moreover, with prob. $\geq 1 - \delta$, any $\widehat{\theta}$ with $\widehat{\mathcal{L}}_{\text{align}}(\widehat{\theta}) \leq \widehat{\eta}$ and $\|\nabla\widehat{\mathcal{L}}_{\text{align}}(\widehat{\theta})\| \leq \varepsilon$ satisfies the same bounds with $\eta = \widehat{\eta} + \mathfrak{R}_n + O(\varepsilon)$, $\mathfrak{R}_n = O\big(\sqrt{\log(1/\delta)/n}\big)$.*

*Proof.* On $\mathbb{S}^{d-1}$, $D(u, v) = 1 - \langle u, v \rangle$. The population InfoNCE risk for batch $B$, temperature $\tau$ is

$$\mathcal{L}_{\text{align}}(\theta) = \mathbb{E}\left[ -\log \frac{e^{\langle s, s^+ \rangle/\tau}}{e^{\langle s, s^+ \rangle/\tau} + \sum_{j=1}^{B-1} e^{\langle s, s_j^- \rangle/\tau}} \right]. \tag{5}$$

For any $a, b_1, \ldots, b_m \in \mathbb{R}$ and $\tau > 0$, the Softmax–margin inequality (Saunshi et al., 2019) is

$$-\log \frac{e^{a/\tau}}{e^{a/\tau} + \sum_{j=1}^{m} e^{b_j/\tau}} \leq \frac{1}{\tau} \max_j (b_j - a) + \log(1 + m). \tag{6}$$

(Alignment, contraction of low entropy segments). Apply equation 6 to equation 5 with $a = \langle s, \mu_c \rangle$, $b_j = \langle s, \mu_c^{-(j)} \rangle$ to obtain $\mathbb{E}\langle s, \mu_c \rangle - \mathbb{E}\langle s, \mu_c^- \rangle \geq \tau(\log B - \eta)$. By Assumption 1 and $\|u - \mu\|_2^2 = 2(1 - \langle u, \mu \rangle)$,

$$\mathbb{E}_r \|s(r, h_c) - \mu_c\|_2^2 \leq \frac{2}{\Delta_{\text{pos}}} \psi_1(\eta, \tau, B). \tag{7}$$

(Uniformity, repulsion of high entropy segments). Set $a = \langle s,\ v(r, h_c) \rangle$, $b_j = \langle s,\ v(r_j, h_c) \rangle$; then $\mathbb{E}\langle s,\ v(r', h_c) \rangle - \mathbb{E}\langle s,\ v(r, h_c) \rangle \geq \tau(\log B - \eta)$. Assumption 1 yields

$$\log \mathbb{E}_{r \neq r'} \exp\big(-t\|s(r, h_c) - s(r', h_c)\|_2^2\big) \geq \frac{1}{\Delta_{\text{neg}}} \psi_2(\eta, \tau, B). \tag{8}$$

For finite-sample, uniform convergence (Saunshi et al., 2019) gives

$$\sup_\theta \left| \widehat{\mathcal{L}}_{\text{align}}(\theta) - \mathcal{L}_{\text{align}}(\theta) \right| \leq \mathfrak{R}_n = O\Big(\sqrt{\frac{\log(1/\delta)}{n}}\Big) \quad \text{w.p.} \geq 1 - \delta. \tag{9}$$

If $\widehat{\mathcal{L}}_{\text{align}}(\widehat{\theta}) \leq \widehat{\eta}$ and $\|\nabla\widehat{\mathcal{L}}_{\text{align}}(\widehat{\theta})\| \leq \varepsilon$, smoothness implies

$$\mathcal{L}_{\text{align}}(\widehat{\theta}) \leq \widehat{\eta} + \mathfrak{R}_n + O(\varepsilon). \tag{10}$$

Set $\tilde{\eta} := \widehat{\eta} + \mathfrak{R}_n + O(\varepsilon)$ and substitute $\eta = \tilde{\eta}$ into equation 7 and equation 8. Routing by $\widehat{H}(c)$ and Assumption 2 give a misrouting probability $\delta_{\text{ent}}$ that vanishes with $m_c$, so the bounds hold with prob. $\geq 1 - \delta - \delta_{\text{ent}}$. $\qquad\square$

**Corollary 1** (Entropy-aware Alignment $\Rightarrow$ Domain Consistency). *With probability at least $1 - \delta - \delta_{\mathrm{ent}}$, if $\eta$ is small and $B$ is large, then*

$$\mathcal{L}_{\mathrm{align}}^{\mathrm{low}}(f; \alpha) \ \leq \ \widetilde{O}\Big(\tfrac{\tau}{\Delta_{\mathrm{pos}}B}\Big),$$

*so low-entropy columns contract toward their semantic centroids, ensuring **Domain Consistency**.*

**Corollary 2** (Entropy-aware Uniformity $\Rightarrow$ Domain Fidelity). *Under the same conditions,*

$$\mathcal{L}_{\mathrm{unif}}^{\mathrm{high}}(f; t) \ \geq \ \widetilde{\Omega}\Big(\tfrac{1}{\tau}\Delta_{\mathrm{neg}}\Big),$$

*so high-entropy columns preserve row-level separation, ensuring **Domain Fidelity**.*

*Remark.* Taken together, these corollaries formalize the *domain manifold geometry* induced by entropy-driven segment alignment: low-entropy (schema-stable) segments contract onto a shared cross-table manifold, while high-entropy (entity-specific) segments distribute across this manifold with entropy-aware uniformity. This structure preserves domain-level coherence while maintaining row-level discriminativity. Here $\widetilde{O}(\cdot)$ and $\widetilde{\Omega}(\cdot)$ suppress polylogarithmic factors in $n$.

# B  RELATED WORKS

## B.1  FIDELITY-ORIENTED ENCODERS

A significant body of research has focused on developing structure-aware encoders, which attempt to explicitly model the 2D layout and relational structure of tables. While foundational, these approaches commonly suffer from two major drawbacks; Inefficiency and Inconsistency.

Inefficiency arises from the architectural complexity required to capture structural cues. These models often introduce significant computational and training overhead. TAPAS Herzig et al. (2020), for example, employs a multitude of embedding layers to encode token roles (e.g., `row_id`, `column_id`, `rank_id`), which is expensive to train. TaBERT Yin et al. (2020) linearizes table content, but its representation is suboptimal; embedding a single cell $\langle i, j \rangle$ requires a minimum of three tokens. For real-world tables with hundreds of columns, this approach quickly becomes infeasible within standard token limits. Other models introduce complexity through architectural choices, such as Tabbie Iida et al. (2021) utilizing two separate transformers, or through intricate encoding schemes. Turl Deng et al. (2022) uses a complex entity representation process involving two role embeddings (type and mention) and a projection layer. Tuta Wang et al. (2021) implements a highly complex positional encoding system with multiple levels of independently learned, tree-based positional encodings, in addition to in-cell positional encodings.

Despite this added complexity, these models fail to achieve robust semantic consistency. Their representations remain vulnerable to simple schema variations, such as column reordering. Furthermore, the embeddings for a given concept can drift semantically depending on the specific query or the context of neighboring entities, indicating a lack of true semantic grounding.

## B.2  CONSISTENCY-ORIENTED ENCODER

More recently, research has shifted toward domain-aware encoders, which prioritize semantic consistency across different table structures, aiming to capture the "domain" of a column. A notable example is HAETAE Jung & Yoon (2025), which contrasts with structure-aware models by using a simpler, lightweight approach. It uses a standard BERT backbone but integrates an additional embedding layer for row context-free header tokens. Haetae trains this universal header embedding using a distance-based objective, which explicitly forces headers with the same semantic meaning (e.g., "First Name" and "f_name") to have similar representations.

While this method successfully ensures header consistency, it introduces a critical limitation: Header-value Misalignment. By forcing header representations to be close while neglecting the semantic information contained in the cell values, the model harms the crucial header-value dependencies. This optimization for header-level consistency weakens the model's ability to perform deep table reasoning. The resulting consistency is not truly grounded in the full domain semantics of the table, as it largely ignores the values, which are essential for defining that domain.

## B.3  TASK-ORIENTED APPROACHES

A line of research focuses on task-specific pretraining, adapting language models to address the heterogeneity of tabular attributes for supervised prediction. TP-BERTa (Yan et al.), for example, is designed explicitly for regression and classification, introducing relative magnitude tokenization and intra-feature attention to reconcile numerical values with feature semantics, thereby competing with strong tree-based and deep tabular baselines. Complementary paradigms expand task awareness in different directions: TAPEX (Liu et al.) pretrains on SQL execution to enhance table QA, while TabPFN (Hollmann et al.) uses synthetic priors for probabilistic classification without finetuning. More recent work pushes toward broader reasoning capabilities, including modular table reasoning with TAPERA (Zhao et al., 2024), instruction-tuned multi-task alignment in Table-GPT (Li et al., 2024), and generative modeling with CDTD (Mueller et al., 2023) for mixed-type imputation. Collectively, these efforts highlight a shift toward tailoring pretraining to specific downstream tasks—whether predictive modeling, QA, or imputation—though such specialization often comes at the cost of limited transferability across domains requiring general-purpose table understanding.

## C  Supplementary Analyses

### C.1  Fidelity under Schema Perturbations

To complement the fidelity analysis in Section 3.2, we further evaluate NAVI's robustness under schema perturbations that mimic realistic inconsistencies in web tables. While the main paper evaluates fidelity on clean schemas, practical deployments must handle lexical drift (e.g., synonyms), noisy or unseen headers (e.g., typos), and structural variation (e.g., column reordering). We therefore apply controlled perturbations at inference time—using models trained strictly on clean schemas—to test whether NAVI preserves both domain fidelity (entity-level discriminability) and structural fidelity (schema–value grounding) under degraded schema conditions.

We consider three perturbation types commonly observed in heterogeneous table corpora:

- **Synonym replacement (semantic OOV).** For each table, we identify low-entropy headers and randomly replace 50% of them with semantically equivalent yet unseen alternatives from a curated synonym mapping (e.g., director.name → auteur.name). This tests whether semantic variants map to the same header manifold region.

- **Header typos (noisy OOV).** We sample 50% of low-entropy headers and apply 1–2 character-level corruptions (substitution, insertion, deletion), producing unseen, noisy forms that break lexical structure. This simulates genuinely OOV headers rather than simple lexical variants.

- **Column reordering (structural noise).** For each row, we randomly permute the column order while preserving header–value associations. This tests whether NAVI's row representations are sensitive to presentation order.

Table 5: F1 Score on Row Classification with XGBoost (domain fidelity) and Accuracy on Value Imputation (structural fidelity) for both Product and Movie domains.

| Model | Product | | Movie | |
|---|---|---|---|---|
| | **R-Cls** (F1) | **Val** (acc.) | **R-Cls** (F1) | **Val** (acc.) |
| Default | 0.9301 | 0.7902 | 0.6394 | 0.7077 |
| Synonym | 0.9466 | 0.7743 | 0.6322 | 0.6918 |
| Typo | 0.9284 | 0.7212 | 0.6103 | 0.6620 |
| Column Reordered | 0.9404 | 0.7830 | 0.6161 | 0.7007 |

Synonym replacement yields performance essentially matching or slightly exceeding the clean-schema baseline (e.g., Product Cls: $0.9295 \rightarrow 0.9466$), indicating that NAVI's global header encoder effectively absorbs semantic variants and maps them to the same low-entropy centroids. This aligns with our consistency analysis: synonym-level drift has minimal effect on the semantic manifold structure. Header typos produce the largest degradation (e.g., Movie Cls: $0.6394 \rightarrow 0.6103$), as expected when character corruption disrupts subword tokenization and weakens header–value grounding. Nonetheless, NAVI retains a substantial portion of its clean-schema fidelity—far from catastrophic failure—demonstrating that entropy-aware alignment provides robust anchoring even under noisy OOV headers. Column reordering results in only minor changes, with performance consistently close to the clean baseline (e.g., Prod Imp: $0.7857 \rightarrow 0.7830$). This confirms NAVI's row-as-multiset design: segment-based representations are largely insensitive to column order, preserving both domain and structural fidelity under presentation-level variability.

Overall, these results show that NAVI's fidelity is highly stable under realistic schema inconsistencies. Semantic variations (synonyms) are effectively normalized; structural perturbations (reordering) have negligible impact; and even noisy OOV cases (typos) degrade performance gracefully rather than collapsing schema–value grounding. These findings reinforce that entropy-aware alignment yields a durable, schema-robust representation of table semantics.

## C.2 GEOMETRY OF SEGMENT EMBEDDINGS

To further examine how NAVI organizes segment-level representations, we visualize segments from five Movie tables using t-SNE, grouping segments by their entropy category (low vs. high) and overlaying table-wise convex hulls (Figure 5). The resulting geometry provides qualitative evidence that NAVI realizes cross-table generalization (*domain consistency*, *domain fidelity*). For BERT, segment embeddings are constructed by mean-pooling the contextualized token embeddings of each header–value span, ensuring a comparable segment representation across models.

Figure 5: t-SNE projections of header–value segment embeddings from five Movie tables, grouped by entropy category. Gray convex hulls correspond to individual tables. For low entropy segments, points are additionally labeled as Best Rating or Worst Rating.

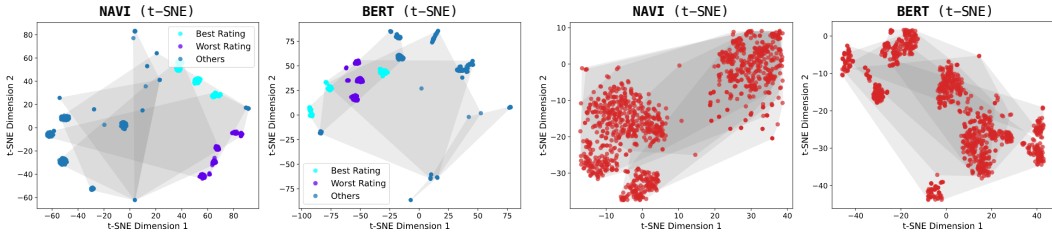

(a) Low entropy segment embeddings.  (b) Segment distribution.

Low-entropy segments correspond to stable domain concepts (e.g., ratings, director). To assess whether models recover these semantics, we color segments using two coherent groups—Best Rating and Worst Rating. Under NAVI, the two groups form clean, well-separated clusters that persist across tables, with overlapping convex hulls indicating that the geometry is driven by shared cross-table value distributions rather than table identity. This reflects strong semantic discrimination and domain-level invariance. Quantitatively, NAVI attains a higher silhouette score (0.7061) than BERT (0.2327), confirming that BERT's clusters remain overlapping and weakly delineated, dominated by table-specific structure. Overall, NAVI collapses surface-form variation while preserving core domain distinctions, yielding representations that generalize consistently across heterogeneous tables.

High-entropy segments represent entity-specific content (e.g., names, descriptions). NAVI distributes these segments broadly, avoiding collapse even when within-table contexts are similar. The table hulls heavily overlap, showing that representations do not cling to table identity. BERT, however, forms several dense, table-specific clumps, indicating that its contextual embedding remains sensitive to table-local patterns and fails to maintain row-level separability across tables. NAVI's geometry thus reflects stronger domain fidelity: entity-specific values remain distinguishable without being entangled with schema or table-specific quirks.

# D IMPLEMENTATION DETAILS

## D.1 ARCHITECTURAL DETAILS

**Global Header Encoder**   The header encoder is implemented as a lightweight BERT-based module that generates context-independent embeddings for header strings. The encoder utilizes BERT tokenizer and embedding layer, followed by two transformer layers (layers 8 and 9 from the pretrained BERT model) to capture semantic representations of header names, effectively leveraging BERT's semantic priors.

The design choice of using two layers strikes a balance between expressivity and efficiency: a shallow encoder reduces computational overhead while still allowing non-trivial contextualization beyond the embedding layer. Using more layers risks overfitting to sentence-level semantics irrelevant for headers, while fewer layers (e.g., only one) limit the ability to model compositional structure.

The selection of layers 8 and 9 is grounded in empirical analysis of BERT (Clark et al., 2019) shows that mid-to-deep layers (approximately layers 7–10) specialize in syntactic dependencies and head–dependent relations, such as determiners linking to nouns and direct objects linking to verbs. By contrast, earlier layers capture mostly local or lexical information, while the final layers (11–12)

are biased toward [CLS]-based sentence aggregation and task-specific adaptation. Leveraging layers 8 and 9 thus provides a strong inductive bias for modeling headers, which are typically short noun phrases requiring syntactic but not full discourse-level context.

Given a header string $h$, the encoder first tokenizes it using the BERT tokenizer, then processes the tokens through the embedding layer to obtain initial representations. These embeddings are passed through two sequential transformer layers with self-attention mechanisms:

$$\mathbf{h}^{(0)} = \text{BertEmbeddings}(\text{tokenize}(h))$$
$$\mathbf{h}^{(1)} = \text{EncoderLayer}_8(\mathbf{h}^{(0)})$$
$$\mathbf{h}^{(2)} = \text{EncoderLayer}_9(\mathbf{h}^{(1)})$$

The final universal header embedding $E_{\text{global}}(h)$ is obtained through mean pooling over the sequence dimension, weighted by the attention mask to exclude padding tokens:

$$E_{\text{global}}(h) = \frac{\sum_{i=1}^n \mathbf{h}_i^{(2)} \cdot \text{mask}_i}{\sum_{i=1}^n \text{mask}_i}.$$

The encoder supports flexible input formats, handling single header strings, flat lists of headers, or batched lists, automatically adjusting the output dimensionality and providing appropriate masking for batch processing.

**Projection Layer for Segments**   The segment projection network $g(\cdot)$ implements the transformation that combines universal header embeddings, contextualized header representations, and contextualized value representations into unified segment embeddings. Motivated by projection layers in transformer-based language models (Vaswani et al., 2017), the architecture adopts a two-stage feedforward block with residual connections and normalization, enabling non-linear feature mixing while maintaining training stability.

Given the three input components $E_{\text{global}} \in \mathbb{R}^{B \times H \times D}$, $H_{\text{ctx}} \in \mathbb{R}^{B \times H \times D}$, and $V_{\text{ctx}} \in \mathbb{R}^{B \times H \times D}$, the projection first concatenates them along the feature dimension:

$$\mathbf{x}_{\text{concat}} = [E_{\text{global}} \parallel H_{\text{ctx}} \parallel V_{\text{ctx}}] \in \mathbb{R}^{B \times H \times 3D}$$

The concatenated representation is then processed through a two-layer feedforward network with GELU activation and layer normalization:

$$\mathbf{x}_{\text{hidden}} = \text{LayerNorm}(\text{GELU}(\text{Linear}3D \rightarrow 2D(\mathbf{x}_{\text{concat}})))$$
$$s(r, h) = \text{LayerNorm}(\text{Linear}2D \rightarrow D(\text{Dropout}(\mathbf{x}_{\text{hidden}})))$$

This design mirrors the intermediate expansion–compression scheme used in transformers, where increasing dimensionality allows richer interactions between features before reducing back to the model dimension for compatibility. By concatenating schema-level and row-level signals, the projection network learns to fuse global header semantics with local contextual patterns. The residual normalization ensures stable optimization, while the intermediate $2D$ bottleneck provides sufficient capacity to capture complex header–value dependencies.

## D.2   Dataset Preprocessing

Our dataset preprocessing pipeline is designed to optimize the quality and compatibility of tabular data for BERT-based language model training. The preprocessing consists of three main stages: data cleaning, BERT vocabulary validation, and tokenization optimization.

**Data Cleaning and Normalization**   The raw tabular data undergoes several cleaning steps to ensure consistency and quality. First, we flatten nested JSON structures. For example:

```
"actors": [{"name": "allan"}, {"name": "daniel"}] →
"actors.0.name": "allan", "actors.1.name": "daniel"
```

This creates a uniform representation where each row is represented as a flat dictionary of key-value pairs. This flattening process preserves the hierarchical structure through dot-separated keys.

Next, we handle indexed fields that represent repeated attributes. To prevent information overload and maintain computational efficiency, we sample a maximum of 3 indexed fields per field type, prioritizing the first occurrences to maintain data consistency.

**BERT Vocabulary Validation** A critical challenge in training BERT on multilingual tabular data is the model's limited vocabulary coverage for non-English languages. To address this, we implement a BERT vocabulary validation step that filters out tables containing content that cannot be effectively tokenized by the BERT tokenizer.

For each table, we extract meaningful text fields (excluding URLs, pure numbers, and very short strings) and tokenize them using the BERT tokenizer. We calculate the ratio of unknown tokens ([UNK]) to total tokens for each field. Tables where more than 30% of the text fields contain excessive unknown tokens (threshold: 30% UNK ratio) are excluded from training. This filtering ensures that the model trains on data it can meaningfully process, significantly reducing the proportion of uninformative [UNK] tokens during training.

**Tokenization Optimization** Finally, to maximize the utility of the remaining data while respecting BERT's token limit constraints, we implement field-level truncation: Individual fields that exceed 20 tokens are truncated to fit within this limit, preserving the most important information while maintaining field names and separators.

**Preprocessing Statistics** Our preprocessing pipeline processes data from 100 different e-commerce websites across multiple languages and domains. The BERT vocabulary validation step typically filters out 60-70% of rows containing significant non-English content, resulting in a dataset focused on English-language e-commerce data that can be effectively processed by BERT.

The final preprocessed dataset maintains the structural information of the original tables while ensuring compatibility with BERT's tokenization scheme, enabling effective representation learning for tabular data through masked language modeling objectives.

This approach addresses the fundamental challenge of applying English-centric language models to multilingual structured data, ensuring that the training process focuses on content that the model can meaningfully learn from while preserving the rich structural information inherent in tabular data.

### D.3 TRAINING PROCEDURE

**Batch Construction.** For each domain, we organize the 100 tables into stratified batches using a hierarchical grouping strategy. Specifically, tables are grouped into sets of four (25 groups per domain), with all rows in a group merged into a unified dataset. An epoch processes all groups sequentially, with group order shuffled each time while preserving the 4-table grouping for computational efficiency. Within each group, stratified sampling assigns batch slots in proportion to table size: $\text{batch\_count}_{t_i} = \max\left(1, \text{round}\left(n_i/N \times \text{batch\_size}\right)\right)$, for table $t_i$ with $n_i$ rows out of $N$. This procedure balances representation across tables, prevents larger tables from dominating training, and ensures that even small tables contribute consistently to every batch.

**Entropy-based Column Categorization** Following the entropy-based categorization described in Section 2.3, we compute normalized Shannon entropy for each field $f$ in table $t$:

$$H_{\text{norm}}(f) = \frac{-\sum_{v \in V_f} p(v) \log_2 p(v)}{\log_2 |V_f|}$$

where $V_f$ is the set of unique values for field $f$, and $p(v)$ is the probability of value $v$. The categorization uses quartile-based thresholds computed per table:

- Domain-coherent columns $\mathcal{H}_{\text{dom}}$: $H_{\text{norm}}(f) \leq Q_1$,
- Entity-discriminative columns $\mathcal{H}_{\text{ent}}$: $H_{\text{norm}}(f) \geq Q_3$,

where domain-coherent columns represent stable, low-entropy fields capturing global domain semantics (e.g., genre), entity-discriminative columns represent high-entropy fields that vary across rows, capturing instance-specific attributes (e.g., title). This per-table categorization ensures robust field classification regardless of table size or domain characteristics, with minimum guarantees of at least one field per category when possible. The categorization is computed once per combined dataset and used throughout the training of that group.

---

**Algorithm 1** NAVI Training Procedure

---

**Require:** Domain $\mathcal{D}$ with tables $\{t_i\}_{i=1}^{100}$, model $\mathcal{M}_\theta$, alignment weight $\lambda_{\text{align}}$,
  masking configuration MaskCfg
**Ensure:** Trained parameters $\theta^*$
1: Initialize $\theta$, optimizer, gradient scaler
2: **for** epoch $t = 1, \ldots, T$ **do**
3:   Partition 100 tables into groups $\mathcal{G} = \{G_1, \ldots, G_{25}\}, |G_i| = 4$
4:   Shuffle group order
5:   **for** each group $G \in \mathcal{G}$ **do**
6:     **for** each table $t_i \in \mathcal{G}$ **do**
7:       Compute normalized entropy per field: $H_{\text{norm}}(f) = -\sum_{v \in V_f} p(v) \log p(v) / \log |V_f|$
8:       Categorize fields: $\mathcal{H}_{\text{dom}} = \{f : H_{\text{norm}}(f) \leq Q_1\}, \quad \mathcal{H}_{\text{ent}} = \{f : H_{\text{norm}}(f) \geq Q_3\}$
9:       Initialize stratified sampler, $\text{Sampler}(\mathcal{R}_G)$
10:    **end for**
11:    **for** each batch $\mathcal{B} \sim \text{Sampler}(\mathcal{R}_G)$ **do**
12:      Apply masking $\text{Mask}(\cdot; \text{MaskCfg})$: $\tilde{\mathbf{x}}_b, \mathbf{y}_b = \text{mask}(\mathbf{x}_b; \text{MaskCfg})$
13:      Forward (masked): logits, $\mathbf{L}_b = \mathcal{M}_\theta(\tilde{\mathbf{x}}_b)$
14:      Forward (unmasked): embeddings, $\mathbf{E}_b = \mathcal{M}_\theta(\mathbf{x}_b)$
15:      Extract components: $\{E_{\text{global}}, H_{\text{ctx}}, V_{\text{ctx}}\} = \text{extract}(\mathbf{E}_b)$
16:      Segment fusion: $s(r, h) = g\big(E_{\text{global}}(h) \parallel H_{\text{ctx}}(r, h) \parallel V_{\text{ctx}}(r, h)\big)$
17:      Compute losses: $\mathcal{L}_{\text{msm}}^{(b)}, \mathcal{L}_{\text{dom}}^t, \mathcal{L}_{\text{ent}}^t$
18:      Total loss: $\mathcal{L}_{\text{total}}^{(b)} = \mathcal{L}_{\text{msm}}^{(b)} + \lambda_{\text{align}} \cdot \frac{1}{|\mathcal{B}|} \sum_{t \in \mathcal{B}} (\mathcal{L}_{\text{dom}}^t + \mathcal{L}_{\text{ent}}^t)$
19:      Update: $\theta \leftarrow \text{step}(\theta, \mathcal{L}_{\text{total}}^{(b)})$
20:    **end for**
21:  **end for**
22: **end for**
23: **return** $\mathcal{M}_{\theta^*}$

---

**Masking Configuration.** Building on the structure-aware MSM framework in Section 2.2, we define three masking regimes with token budget control: (1) Header–Value (HV) Masking: Selects $k$ header and value segments under a total budget $\frac{\texttt{max\_tokens}}{\texttt{token\_length\_threshold}}$, split by a configurable ratio (default: 50% values, 50% headers), with each segment contributing up to 8 tokens to $\mathcal{M}_h$ or $\mathcal{M}_v$. (2) BERT-style (B) Masking: Standard MLM regime with 15% uniform masking over non-special tokens to form $\mathcal{M}r$. (3) Hybrid (HVB) Masking: Combines the two by allocating $w_{hv} \times \texttt{max\_tokens}$ (default $w_{hv} = 0.5$) to HV masking and the remainder to BERT-style masking. All regimes follow the usual replacement scheme (80% [MASK], 10% random, 10% unchanged).

**Forward Pass and Loss Computation.** The forward pass follows the semantic-aware schema induction framework (Section **??**) and enriched by universal header embeddings described in Appendix D.1. For each batch, the model performs two passes: (1) Masked input $\rightarrow$ MSM logits for structure-aware segment modeling; (2) Unmasked input $\rightarrow$ contextualized embeddings for entropy-aware contrastive alignment. From the unmasked pass, we extract universal header embeddings $E_{\text{global}}(h)$, contextualized header representations $H_{\text{ctx}}(r, h)$, and value representations $V_{\text{ctx}}(r, h)$, which are fused via the projection network $g(\cdot)$ into segment embeddings $s(r, h)$. The total loss combines structure-aware MSM and entropy-aware contrastive alignment: $\mathcal{L}_{\text{total}} = \mathcal{L}_{\text{msm}} + \lambda_{\text{align}} \cdot \mathcal{L}_{\text{align}}$, where $\mathcal{L}_{\text{msm}}$ is computed over the masked sets $\mathcal{M}$, and $\mathcal{L}_{\text{align}} = \mathcal{L}_{\text{dom}}^t + \mathcal{L}_{\text{ent}}^t$ jointly enforces domain consistency (cross-header alignment) and domain fidelity (cross-row alignment).

### D.4 TRAINING REGIMEN AND HYPERPARAMETERS

Figure 6: Learning curves: Batch size 32 for 2 epochs, total number of steps is 21132.

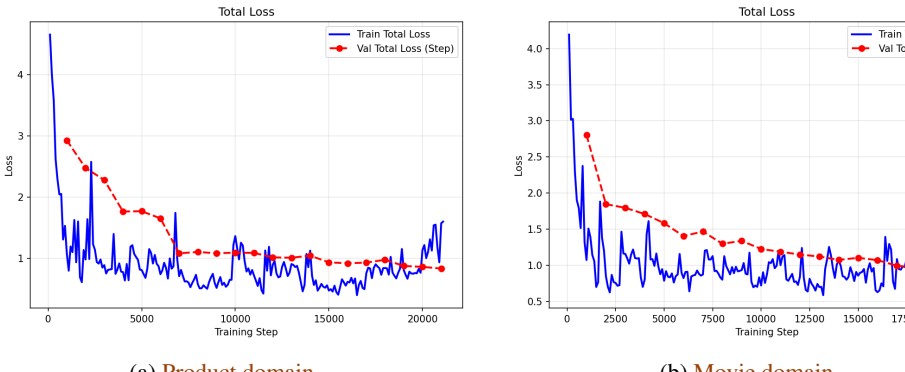

(a) Product domain.  (b) Movie domain.

Table 6: Performance of hyperparameter-tuned variants on downstream fidelity tasks. Evaluation includes Value Imputation (Val; accuracy), Row Classification with XGBoost (XGB; F1-Macro) and Logistic Regression (LR; F1-Macro). Best results per task are in bold.

| | Product | | | Movie | | |
|---|---|---|---|---|---|---|
| | Val | XGB | LR | Val | XGB | LR |
| Default | **0.7902** | 0.9295 | **0.9410** | 0.7077 | **0.6394** | **0.6667** |
| $\lambda_{align}$ | | | | | | |
| 0.01 | 0.7773 | 0.9115 | 0.9298 | 0.7009 | 0.5994 | 0.6495 |
| 0.25 | 0.7795 | 0.7513 | 0.8648 | 0.7023 | 0.5151 | 0.5615 |
| 1.25 | 0.7846 | 0.7888 | 0.9057 | 0.7046 | 0.3970 | 0.4624 |
| $h{:}v{:}b$ | | | | | | |
| 6:2:2 | 0.7782 | 0.9246 | 0.9359 | 0.7028 | 0.5757 | 0.6053 |
| 2:6:2 | 0.7762 | 0.9087 | 0.9379 | **0.7240** | 0.5939 | 0.6312 |
| 3:1:6 | 0.7793 | 0.8971 | 0.9289 | 0.7055 | 0.5798 | 0.6308 |
| 2:2:6 | 0.7779 | 0.9227 | 0.9399 | 0.7036 | 0.5985 | 0.6394 |
| 1:3:6 | 0.7819 | 0.9246 | 0.9388 | 0.6970 | 0.6034 | 0.6439 |
| *entropy thres.* | | | | | | |
| Q1/Q3 | 0.7789 | 0.9188 | 0.9330 | 0.7053 | 0.5043 | 0.5446 |
| 40p/60p | 0.7204 | 0.8056 | 0.8955 | 0.6710 | 0.2236 | 0.3433 |
| 50p | 0.7086 | 0.7805 | 0.8688 | 0.6460 | 0.1852 | 0.2484 |
| $\tau_{dom}/\tau_{ent}$ | | | | | | |
| 0.02/0.02 | 0.7716 | **0.9310** | 0.9399 | 0.7052 | 0.6174 | 0.6442 |
| 0.14/0.14 | 0.7813 | 0.7446 | 0.8829 | 0.7021 | 0.3917 | 0.4963 |
| *negative size* | | | | | | |
| 16 | 0.7750 | 0.9228 | 0.9400 | 0.6888 | 0.5992 | 0.6396 |
| 32 | 0.7565 | 0.8959 | 0.9269 | 0.7069 | 0.5902 | 0.6078 |

