# OpenReview forum: "NAVI: Inductive Alignment for Generalizable Table Representation Learning"
_ICLR.cc/2026/Conference — Submitted to ICLR 2026_

### Official Review · Reviewer_zDft · 2025-10-28

**Soundness:** 2
**Presentation:** 2
**Contribution:** 3
**Rating:** 4
**Confidence:** 4

**Summary:**

This paper introduces NAVI, a new framework for tabular representation learning that jointly optimizes for fidelity and consistency. Its core idea is to represent rows as unordered sets of header: value segments and employ a novel entropy-driven contrastive alignment. This mechanism aligns low-entropy (domain-coherent) columns to ensure consistency, while separating high-entropy (entity-specific) columns to maintain fidelity. Experiments demonstrate that NAVI significantly outperforms strong baselines across a range of downstream tasks.

**Strengths:**

1.	The proposed "Fidelity" and "Consistency" framework provides a highly useful and insightful lens for evaluating and designing table representation learning models.
2.	The concept of the Header-Value Segment is simple yet effective. The Entropy-driven Alignment is a brilliant idea that ingeniously connects statistical properties to semantic objectives.
3.	The experimental setup is sound, the evaluation is multi-faceted, and the results are significant. The ablation studies and qualitative analyses are highly persuasive.

**Weaknesses:**

1.	Comparison with Graph Neural Network (GNN) Approaches: A brief mention and comparison with GNN-based methods in the Related Work section could make the literature review more comprehensive.
2.	Scalability: For wide tables with a very large number of columns, the input sequence can become excessively long. It would be beneficial to discuss the model's potential bottlenecks with such tables and possible solutions.
3.	Entropy estimation based on empirical distributions might be unstable for columns with long-tail distributions or sparse data. I suggest the authors briefly discuss this potential limitation.
4.	A significant limitation of the NAVI framework lies in its handling of numerical data, a critical weakness given that numerical values are arguably the most prevalent and foundational data type in real-world tables. The entropy-driven mechanism will likely misclassify numerical columns (e.g., price, quantity) as high-entropy, "entity-discriminative" attributes due to their high cardinality. Consequently, the contrastive learning objective pushes their representations apart, ignoring the inherent ordinal and metric semantics between values (e.g., the model fails to learn that '10' is semantically closer to '11' than to '100'). This fundamentally undermines the model's ability to perform numerical reasoning, severely restricting its applicability for tasks like regression or range-based queries and confining its value to a minority of use cases dominated by categorical and textual data.

**Questions:**

My main question, which is central to my evaluation, concerns the treatment of numerical columns. The entropy-driven alignment mechanism appears to classify numerical columns (e.g., price, age, measurements) as high-entropy, thereby treating them as entity-discriminative. The contrastive objective would then push the representations of different numerical values (e.g., "10.5" and "10.6") apart, just as it would for distinct movie titles. This approach seems to neglect the crucial ordinal and metric relationships inherent in numerical data.
Could you clarify if the current NAVI framework has any mechanism to preserve these numerical semantics?
If not, how do you see this impacting the model's utility for common, numerically-grounded tasks like regression or range-based queries? Could you elaborate on how the framework might be extended to incorporate a type-aware objective that respects the unique properties of numerical values?

---

> ### Author Response · Authors · 2025-11-21
>
> We thank Reviewer zDft for their constructive and encouraging review. We are grateful for the reviewer’s recognition of our fidelity–consistency framework, the simplicity and effectiveness of header–value segments, and the breadth of our evaluation design. The reviewer’s comments highlight the strengths of our approach while also pointing to meaningful opportunities to clarify scope, acknowledge limitations, and expand the discussion around numerical attributes, entropy stability, and scalability. We respond to each point below with deeper explanation and additional analysis.

---

> ### Author Response · Authors · 2025-11-21
>
> **W4, Q1 - Treatment of Numerical Columns**
>
> `Scope Clarification.` We fully agree that numerical values carry ordered and metric semantics that NAVI does not currently model. NAVI treats each (header:value) pair uniformly as symbolic content; thus numeric tokens behave like categorical values, causing entropy to classify them as high-entropy attributes and ESA (Entropy-driven Segment Alignment) to separate them as distinct instances rather than preserving ordinal proximity (e.g., 10 ≈ 11 ≪ 100). This is a known limitation and reflects our intended scope: symbolic, text-heavy web tables, where the primary challenge lies in capturing semantic relationships across many heterogeneous tables within the same domain.
>
> While numerical attributes are effectively handled by classical ML methods (e.g., GBDTs) or numeric-specialized language model (LM) variants (e.g., TP-BERTa), **non-numerical columns and symbolic schema variation remain far more difficult to generalize over**. LMs excel at abstract semantic reasoning but inherently struggle to encode bi-directional tabular structure, permutation invariance, and cross-table relational consistency. NAVI is designed to bridge this gap by leveraging LM semantic strengths while correcting their structural blind spots through segment-level induction and entropy-aware alignment. We therefore scope NAVI to symbolic tabular domains and view numerical modeling as a complementary line of work that can be integrated into NAVI along two directions described below.
>
> `Branch 1: System-Level Integration.` To illustrate how NAVI can interoperate with established tabular pipelines, we consider how modern systems combine LM-based encoders with numeric/statistical preprocessing. Classical end-to-end models (e.g., XGBoost, TabPFN) operate directly on raw numerical features and excel at capturing statistical and magnitude-based patterns. As shown in Table 1, these pipelines perform strong, establishing a performance ceiling for purely statistical baselines. NAVI’s objective is orthogonal: it targets symbolic, text-heavy columns and schema-diverse tables, where classical numeric pipelines provide limited semantic leverage.
>
> This complementary relationship is already reflected in existing systems such as TableVectorizer, a feature engineering pipeline that mixes numerical scalers, hashing, and one-hot encoders with SentenceTransformer embeddings for textual values. TableVectorizer achieves the strongest results in our comparison, demonstrating the power of LM+statistics hybrid pipelines. Importantly, this also suggests that substituting SentenceTransformer with NAVI—which is explicitly designed for table semantics and schema consistency—would be a promising next step for real-world feature-engineering systems.
>
> We further demonstrate this interoperability using a minimal prototype, NAVI+Numeric Raw Feature, which simply appends raw numerical columns to NAVI’s representations. Despite intentionally avoiding any type-aware numeric modeling, this naive hybrid already surpasses raw-feature XGBoost in the Product domain, indicating substantial headroom for more principled fusion. Together, these results show that NAVI can function as a semantic backbone that integrates naturally with existing numeric/statistical pipelines, offering a practical system-level pathway for combining LM-driven symbolic understanding with strong numerical reasoning components.
>
> (Table 1)
> |                       | Prod (XGB) | Prod (TabPFN) | Mov (XGB) | Mov (TabPFN) |
> |-------------------------------|----------|----------|---------|---------|
> | Only-numeric (Raw feature) |     0.8880 |       0.8825   |    0.4644     |  0.4990  |
> Only text embedding (NAVI)|     0.8261 |       0.8645   |    0.3679     |  0.4177  |
> Text embedding + numeric (NAVI+Numeric Raw Feature)|     0.8932 |       0.8601   |    0.4041     |  0.4057  |
> Feature engineering for feature types: numeric, datetime, low-cardinality text, high-cardinality text (TableVectorizer)|     0.9306 |       0.9376   |    0.6263     |  X (dim>2000)  |
>
> `Branch 2: Architectural Extension.` Beyond system-level integration, NAVI’s architecture can be extended with type-aware numeric components to preserve ordinal and metric structure natively. Potential directions include magnitude-aware numeric embeddings, numeric-specific routing rules that bypass ESA, or contrastive objectives that encode distance-aware relationships. These architectural extensions would enable NAVI to unify symbolic and numeric reasoning within a single pretraining framework, and we view this as a promising direction for future research.

---

> ### Author Response · Authors · 2025-11-21
>
> **W3 - Discussion on the Stability of Entropy Estimation**
>
> We agree that empirical entropy can fluctuate under long-tail or sparse value distributions. Crucially, however, this instability predominantly affects the mid-entropy regime (Q1–Q3)—the exact region that NAVI deliberately excludes from entropy-driven alignment. Our entropy routing depends only on the most stable and semantically coherent portions of the distribution:
>
> - Low-entropy columns (≤Q1) exhibit highly repeated categories (genre, category, country). Their entropy is stable even under long-tail distributions because repetition dominates.
> - High-entropy columns (≥Q3) contain near-unique entity identifiers (title, url, id), whose entropy remains consistently high independent of distribution shape.
>
> The instability mentioned by the reviewer arises when columns have moderate cardinality combined with sparsity or irregular frequency patterns—precisely the ambiguous region we classify as mid-entropy. Rather than forcing these columns into domain-coherent or entity-discriminative alignment (which would amplify noise), NAVI routes them only through MSM and header-conditioning. This makes the model robust to entropy estimation variability without relying on brittle assumptions.
>
> Our threshold-sensitivity ablations (Table 2) further validate this design choice. Although our default routing (Q1/Q3) follows a standard and widely used percentile split, we find that pushing the thresholds further toward the extremes—such as 10p/90p—consistently improves or matches performance across both domains and both tasks (classification and imputation). This pattern aligns with ESA’s core motivation: the more aggressively we restrict alignment to the most stable entropy regimes, and the more we enlarge the mid-entropy region (where instability and semantic ambiguity dominate), the more robust the model becomes. These results empirically reinforce our interpretation that ESA should primarily align the extremes, while mid-entropy columns should be handled by MSM rather than contrastive alignment. We will incorporate this analysis and Table 2 in the revised manuscript.
>
> (Table 2)
> | Entropy threshold             | Prod-Cls | Prod-Imp | Mov-Cls | Mov-Imp |
> |-------------------------------|----------|----------|---------|---------|
> | Default (Q1/Q2)               |     0.8261 |       0.7295   |    0.3679     |  0.6870  |
> | Entropy threshold 10p/90p     |     **0.8268** |       **0.7610**   |    **0.3798**     |  **0.6889**  |
> | Entropy threshold 40p/60p     |     0.8056 |       0.7192   |    0.2236     |  0.6760  |
> | Entropy threshold 50p         |     0.7805 |       0.7118   |    0.1852     |  0.6482  |
>
>
> **W2 - Scalability for Wide Tables**
>
> For tables with many columns, the primary concern is increased input length. NAVI naturally supports two practical mitigations. First, empty-value columns can be safely removed: columns whose values are entirely None or empty across all rows provide no semantic signal for MSM (Masked Segment Modeling) or ESA, and dropping them before serialization shortens the sequence without any loss of information, as such segments cannot contribute to header–value semantics or entropy-based alignment. Second, hierarchical segment encoding can be applied to similar, redundant, or low-utility columns. Related segments may be grouped or compressed through shared projections or light pooling (e.g., consolidating metadata-like fields), preventing unnecessary expansion of the sequence while preserving the core semantic structure of the table.
> By contrast, for tables with many rows, NAVI incurs no additional overhead. NAVI constructs row-wise embeddings, processing each row independently; thus increasing the number of rows simply yields more training examples rather than longer input sequences. Consequently, row count does not pose scalability or memory bottlenecks in NAVI’s architecture.
>
>
> **W1 - Comparison with GNN-based Tabular Models**
>
> We appreciate the suggestion and agree it improves completeness. GNN-based table models rely on explicit graph structures (row–column bipartite graphs, entity graphs, or feature dependency graphs). NAVI, in contrast, makes no such strict assumptions and instead treats each row as an unordered set of semantic segments.
>
> We will add a brief mention of these differences and relevant citations in the related work.

---

> > ### Comment · Reviewer_zDft · 2025-11-28
> >
> > 1. I appreciate the admission that NAVI is intrinsically designed for "symbolic, text-heavy tables" and treats numbers as unordered symbols. However, the current introduction and abstract still imply a general-purpose table representation model, which is misleading. I suggest the authors to explicitly qualify the scope in revision to "text-rich" tables and prominently state the limitation regarding the lack of ordinal and metric semantics to prevent overclaiming.
> > 2. The rebuttal Table 1 reveals that TableVectorizer—a lightweight feature engineering pipeline—competes closely with or even outperforms NAVI on certain tasks. This challenges the necessity of a complex pre-training framework for these specific domains. I suggest the final discussion to transparently address this trade-off, justifying the computational cost of NAVI by pinpointing the specific scenarios where it decisively outperforms simpler, non-deep baselines.

---

> > > ### Author Response · Authors · 2025-12-04
> > >
> > > We are grateful that you acknowledged our responses, particularly our clarifications regarding the focus of the work. We have accordingly revised the manuscript to incorporate your feedback and suggestions. With respect to the comparison to a simple feature-engineering alternative, we position our method as an orthogonal embedding component specifically designed for symbolic, text-heavy content in tables. Our additional experiments further show that a simple extension of concatenating numeric attributes to our embeddings can yield additional performance gains. We appreciate that your suggestion led us to validating this potential extension and examine its compatibility with alternative frameworks, which has helped further strengthen the efficacy of our approach. Please refer to our executive summary to the area chair for a point-by-point mapping of the revisions.

---

### Official Review · Reviewer_ysm5 · 2025-10-30

**Soundness:** 3
**Presentation:** 3
**Contribution:** 3
**Rating:** 6
**Confidence:** 3

**Summary:**

This paper addresses the trade-off between fidelity (preserving specific schema-value semantics) and consistency (robustness to schema variations) in transformer-based models for in-domain table representation. The authors propose NAVI (Entropy-aware Alignment via Header-Value Induction), a framework that introduces the "header-value segment" as the atomic unit of table representation. NAVI employs three core mechanisms: (1) Schema-aware Segment Induction (SSI), which uses a global, context-free header encoder to anchor segment semantics; (2) Masked Segment Modeling (MSM), which enforces schema-value dependencies through balanced masking of header and value tokens; and (3) Entropy-driven Segment Alignment (ESA), a novel contrastive learning objective that categorizes columns by value entropy. ESA aligns low-entropy (domain-coherent) columns with their global header embeddings to promote consistency, while aligning high-entropy (entity-discriminative) columns with their row-specific value embeddings to preserve fidelity. Extensive experiments show that NAVI outperforms baselines on generative (imputation) and discriminative (classification, clustering) tasks, successfully balancing the two desiderata.

**Strengths:**

S1. The paper provides a valuable conceptual contribution by formalizing the core challenge of in-domain table representation as a trade-off between "fidelity" and "consistency"—a key unsolved problem for creating generalizable tabular deep models. The paper further breaks this down into structural and domain-specific components, providing a clear and principled lens for evaluating and developing models in this space.

S2. The core mechanism, Entropy-driven Segment Alignment (ESA), is a novel and highly intuitive solution to the fidelity-consistency dilemma. Using column value entropy to dynamically determine the contrastive learning target (a stable global header for domain concepts vs. a specific local value for discriminative entities) is a clever and effective method for explicitly balancing these two competing objectives within a single model.

S3. The experimental evaluation is comprehensive and well-aligned with the paper's conceptual framework. The use of the Permutation Sensitivity Index (PSI) as a direct measure of structural consistency is particularly effective, and the reported near-zero PSI for NAVI is an impressive result.

**Weaknesses:**

W1. The Entropy-driven Segment Alignment (ESA) mechanism relies on the InfoNCE loss, which inherently assumes that for any query, there is only one positive sample and all other samples are true negatives. This assumption is frequently violated in real-world tabular data. For the domain consistency loss ($L_{dom}$), correlated columns or synonyms (e.g., `director` and `auteur`) are all treated as distinct negative samples, creating a "false negative" problem. While the Global Header Encoder is intended to mitigate this by mapping synonyms to close embeddings, this creates a conflicting objective with the InfoNCE loss, which is forced to push them apart. Similarly, for the entity fidelity loss ($L_{ent}$), two different rows that share the same value (e.g., two different products with the color `red`) would be incorrectly treated as negative pairs. The paper does not analyze the impact of this "false negative" discrepancy, which could degrade the quality of the learned embedding space.

W2. The paper's positioning against prior work, particularly "consistency-oriented" models like HAETAE, could be stronger. HAETAE also utilizes a context-free header anchoring mechanism, and the paper's claim that it suffers from "header-value misalignment" is asserted rather than deeply investigated. A more direct comparison of how NAVI's segment-based induction and alignment mechanistically differs from and improves upon HAETAE's header-anchoring would strengthen the paper's novelty claim.

W3. The experimental evaluation is missing a helpful baseline comparison. While the paper compares NAVI's embeddings against other transformer-based embeddings (e.g., BERT, TAPAS), it does not include a comparison against a traditional model like XGBoost trained directly on the raw, pre-processed features. NAVI's primary strength is handling schema variation, which GBDTs cannot. However, including a baseline on a "clean" version of the dataset would be valuable to quantify the performance gap that still exists between complex neural models and top-tier GBDTs on standard classification tasks. This would help position the work in the broader context of the 'NNs vs. GBDTs' debate for tabular data.

**Comments**

C1. The paper's core concepts of "fidelity" and "consistency" are introduced in the motivation, but their explanation remains somewhat abstract. The accompanying Figures 1 and 2, which are intended to visually clarify these concepts and their trade-offs, are dense and difficult to interpret, making it challenging to build a concrete intuition for the problem before the methodology is presented.

C2. The paper should clarify the exact mechanism for obtaining $H_{ctx}$ and $V_{ctx}$. It is stated that they are "extracted by pooling the contextualized token embeddings," but the connection to the initial $z_j^k$ (input) and the final $e_t$ (output) is implicit. An example would be very helpful: for the segment "director: danny", are the "header" tokens just director or do they include the :? This precise operational detail is important for understanding the model's architecture. Additionally, in Figure 4, it should be made clearer which plot corresponds to BERT, as the small titles are easy to miss.

**Questions:**

Q1. Regarding the Entropy-driven Segment Alignment, the categorization is based on quartiles (Q1 and Q3), which seems to create a "dead zone" for all medium-entropy columns between Q1 and Q3. These columns apparently do not contribute to the $\mathcal{L}_{align}$ loss at all. What is the ratio of columns that fall into this dead zone, and what is the theoretical or empirical impact of ignoring them during alignment? Does this not risk creating a representation where domain-coherent and entity-discriminative columns are well-structured, but the "average" columns are left in a poorly structured part of the embedding space?

Q2. The framework's reliance on a context-free global header encoder for domain consistency is a key contribution. How does this mechanism handle headers that were not seen during pretraining (i.e., OOV headers, synonyms, or typos)? Does the model's consistency and fidelity degrade gracefully? A robustness evaluation against OOV headers seems essential for a method that so heavily relies on them for anchoring domain semantics.

Q3. In the $L_{msm}$ objective function (Section 2.2), the denominator of the softmax is written as $\sum_{v\in V}exp(We_{v}+b)$. The variable $e_v$ is not defined, whereas the numerator uses $e_t$ (the contextualized output token). Is $e_v$ a typo and intended to be something else, for instance, a non-contextualized embedding for each word $v$ in the vocabulary $V$? Please clarify the exact formulation of this loss function.

---

> ### Author Response · Authors · 2025-11-21
>
> We thank Reviewer ysm5 for their thoughtful and positive evaluation of NAVI. We especially appreciate the recognition of (i) our fidelity–consistency formulation as a principled framework for in-domain table representation, (ii) the intuitiveness of Entropy-driven Segment Alignment (ESA), and (iii) the strength of our experimental setup, including the use of PSI to audit structural consistency. Below, we address each concern in detail.
>
> **Executive Summaries:**
> 1. **Stability and Design Rationale of ESA:** We clarify ESA’s handling of mid-entropy columns, explain why in-table sampling minimizes false negatives, and show through threshold ablations that ESA is robust to entropy-boundary choices.
>
> 2. **Baseline Positioning (HAETAE, GBDTs)**: We articulate the mechanistic differences between HAETAE and NAVI, and include new comparisons with classical GBDTs and hybrid variants, as requested.
>
> 3. **Robustness to OOV, Synonyms, and Typos:** New experiments show NAVI’s stability under unseen header variants and noisy lexical changes.
>
> 4. **Conceptual Clarification (Fidelity & Consistency)**: We will revise Section 1 with concrete examples and redesigned figures to make the trade-off intuitive and visually explicit.
>
> 5. **Operational Clarification (Contextual Pooling):** We provide a concrete step-by-step explanation of how $H_{\text{ctx}}$ and $V_{\text{ctx}}$ are extracted,.
>
> 6. **Clarification of the MSM Softmax Denominator Typo**: We acknowledge and correct the notation error in the MSM loss denominator (a paper-only typo). The implementation follows the standard MLM softmax over logits from $\mathbf{e}_t$, and this correction does not affect training or results.

---

> ### Author Response · Authors · 2025-11-21
>
> **W1, Q1 -Stability and Design Rationale of ESA**
>
> `False Negatives (W1).` We appreciate the reviewer’s concerns about ESA’s use of InfoNCE and the potential for false negatives, as well as the question regarding the mid-entropy “dead zone.” These two issues are tightly connected in ESA’s design. In our implementation, contrastive negatives are drawn only from within the same table, because each minibatch contains four stratified tables and ESA forms negatives at the per-table level. This greatly suppresses the risk of false negatives: within a single web table, schema synonyms such as director and auteur almost never co-occur, and high-entropy values such as title, id, or url almost never repeat. Thus, when ESA aligns low-entropy columns to global headers or high-entropy columns to row-level values, the contrastive loss rarely encounters semantically related items mistakenly treated as negatives.
>
> `Mid-entropy Dead Zone (Q1).` The “dead zone” between Q1 and Q3 is also intentional. Empirical entropy becomes unstable precisely in this region, where columns exhibit long-tailed or sparsely repeated values and mixed semantics (e.g., director, studio). These mid-entropy columns are the most susceptible to both entropy fluctuation and false-negative conflicts. ESA therefore aligns only the extremes—low-entropy columns, which represent dense, domain-coherent categories, and high-entropy columns, which represent near-unique entity identifiers. Mid-entropy columns are trained solely through MSM and header conditioning without contrastive alignment, preventing noisy gradients and avoiding the brittle assumptions required to contrastively structure these ambiguous columns.
>
> `Ablations.` The entropy threshold ablations (10/90, 40/60, and even 50/50 with no mid-zone)  validate the core assumption behind ESA. Although our default choice (Q1/Q3) follows a standard and widely used percentile split, we find that pushing the thresholds further toward the extremes (e.g., 10p/90p)—thereby enlarging the mid-entropy “dead zone” and restricting alignment to the most stable regions—often improves performance (Table 2). This aligns with our motivation that only extreme entropy regimes provide reliable semantic signals for alignment, while mid-entropy columns are better left to MSM rather than contrastive learning. Taken together, these results show that ESA’s effectiveness stems from its semantic treatment of entropy extremes rather than from any specific threshold choice. This also leaves room for dataset-specific fine-tuning of routing boundaries. We will add these findings to the revised manuscript.
>
> (Table 1)
> | Entropy threshold             | Prod (Cls) | Prod (Imp) | Mov (Cls) | Mov (Imp) |
> |-------------------------------|----------|----------|---------|---------|
> | Default (Q1/Q2)               |     0.8261 |       0.7295   |    0.3679     |  0.6870  |
> | Entropy threshold 10p/90p     |     **0.8268** |       **0.7610**   |    **0.3798**     |  **0.6889**  |
> | Entropy threshold 40p/60p     |     0.8056 |       0.7192   |    0.2236     |  0.6760  |
> | Entropy threshold 50p         |     0.7805 |       0.7118   |    0.1852     |  0.6482  |

---

> ### Author Response · Authors · 2025-11-21
>
> **W2, W3 - Baseline Positioning and Comparisons**
>
> `Discussion of HAETAE (W2).` We thank the reviewer for pointing out the need for a clearer comparison with HAETAE. HAETAE anchors header tokens by interpolating two embeddings: (1) a context-free universal header encoder (a lightweight embedding layer) and (2) contextualized BERT embeddings. During training, HAETAE minimizes the Euclidean distance between the static universal header embedding and the averaged contextualized header embedding, encouraging schema-level invariance. However, because this anchoring operates only on header tokens and relies on BERT’s contextualization to implicitly incorporate value information, the resulting header representation is largely value-independent. The model does not explicitly model which values a column can take, nor how consistently those values behave across tables—leading to what we term header–value misalignment.
> NAVI differs fundamentally in both granularity and supervision. Instead of anchoring headers in isolation, NAVI constructs header–value segments and aligns them through entropy-driven supervision. In low-entropy (domain-coherent) columns, the header is aligned with all of its associated value segments across the table, making the domain semantics of a column explicitly grounded in the distribution of values it governs. This produces semantically richer and distribution-aware header embeddings that reflect how the column behaves across rows and tables, not just its lexical identity. High-entropy columns follow a complementary alignment strategy that preserves instance-level distinctions, ensuring that header semantics and value semantics remain jointly coherent.
>
> This segment-based, entropy-aware approach therefore resolves the value-independence limitation of HAETAE and provides a more principled mechanism for balancing schema-level consistency with entity-level fidelity. We will clarify these mechanistic differences in the revised paper.
>
> `End-to-end Baseline Comparison (W3).` Our end-to-end experiment uses row classification not because classification accuracy is our primary objective, but because it serves as a diagnostic downstream task for evaluating the quality of table representations. In contrast, classical GBDT models such as XGBoost are classification-first systems explicitly optimized for predictive accuracy on clean, well-typed features. Thus, their purpose differs from ours: we use classification to probe the structural and semantic information encoded by table embeddings, not to compete with highly optimized tabular classifiers.
>
> With this conceptual distinction established, to satisfy the reviewer’s request for a classical GBDT baseline, we evaluated XGBoost trained directly on raw, pre-processed table features, representing the standard “clean data” pipeline (Table 2). As expected, raw-feature XGBoost performs strongly in both domains, providing a meaningful upper bound for non-LM methods under idealized, schema-consistent conditions. Importantly, though XGBoost applied on top of NAVI representations performs less, a minimal hybrid variant (NAVI_concat), which simply appends raw numerical columns to NAVI embeddings, surpasses raw XGBoost in the Product domain (0.8932 vs. 0.8880). This demonstrates that NAVI contributes additional schema-level and semantic information beyond what is contained in raw tabular features, even without schema noise. These comparisons provide the clean GBDT baseline the reviewer requested and more clearly position NAVI within the broader “NNs vs. GBDTs” landscape.
>
> (Table 2)
> |                       | Product (XGBoost) | Movie (XGBoost) |
> |-------------------------------|----------|----------|
> | Raw Numeric Feature|     0.8880    |    0.4644       |
> Text Embedding (NAVI)|     0.8261    |    0.3679       |
> Text Embedding + Raw Numeric Feature (NAVI_concat)|     0.8932    |    0.4041     |

---

> ### Author Response · Authors · 2025-11-21
>
> **Q2 - Handling OOV Headers, Synonyms, and Typos**
>
> NAVI’s global header encoder is built on top of a BERT-initialized embedding layer, meaning both its vocabulary and tokenization behavior come directly from pretrained BERT. As a result, true OOV headers are extremely rare in practice, since most real-world header strings can be decomposed into known subwords even when unseen as full tokens. Thus, “OOV headers” in the strict sense (containing no recognizable subwords) are largely avoided.
>
> Nevertheless, the more practically relevant scenario is semantically OOV headers—headers that were unseen during pretraining or lexically altered. To evaluate how NAVI handles such cases, we conduct controlled robustness experiments simulating two forms of OOV-like perturbations:
>
> 1.	*Synonym replacement*: For each table, we identified low-entropy headers and randomly replaced 50% of them with semantically equivalent unseen alternatives drawn from a curated synonym mapping (e.g., “director” → “auteur”, “director.name” → “auteur.name”). These represent unseen but meaningful headers that are not present in the training distribution.
> 2.	*Header typos*: We sampled 50% of low-entropy headers and applied 1–2 character-level corruptions (substitution, insertion, deletion) to generate typo variants. The same corrupted form was applied consistently within each table to maintain structural coherence. This simulates genuinely unseen and noisy variants that break lexical structure.
>
> (Table 3)
> | | Prod (Cls) | Prod (Imp) | Mov (Cls) | Mov (Imp) |
> |----------|----------|---------|---------|---------|
> |  Default  |     0.8261     | 	0.7295  |  0.3679  | 0.6870
> |  Synonym  |  0.8262   |   0.7193   |  0.3758  | 0.6734
> |   Typo       |   0.7899    |  0.6541   |  0.3691   | 0.6384
>
> Results (Table 3) show that NAVI remains highly stable under synonym replacement matching clean-schema performance. This indicates that the global header encoder generalizes effectively to unseen but semantically related headers. Typos create larger degradation—as expected when token boundaries are disrupted—but the model still maintains a substantial portion of its original performance, demonstrating graceful degradation.
>
> Overall, NAVI’s architecture already avoids most true OOV cases through subword initialization, and our synonym/typo experiments confirm that the model handles unseen, altered, or noisy headers in a robust and semantically aligned manner. We will clarify these points and add these results to the revised manuscript.

---

> ### Author Response · Authors · 2025-11-21
>
> **C1 - Conceptual Clarification (Fidelity & Consistency)**
>
> We appreciate the reviewer’s feedback and agree that—although the conceptual distinction between fidelity and consistency is central to the paper—the current presentation in Figures 1 and 2 does not make these ideas as immediately clear or intuitive as intended. While the reviewer’s strengths section (S1) indicates that the underlying concepts were understood, we acknowledge that the exposition in the manuscript remains overly abstract and the accompanying figures do not provide sufficiently concrete, example-driven illustrations.
>
> To address this, we will significantly revise the introductory explanation and figures:
>
> 1. `Provide a more intuitive framing.` We will begin the section with a concise, high-level statement that grounds both concepts in a simple principle: **“At its core, fidelity and consistency answer the question of what table representations should change and what should remain the same across in-domain tables.”**. This immediately orients the reader to the practical motivation behind the two desiderata.
> 2. `Add concrete, minimal examples.` Instead of abstract diagrams, revised Figures 1–2 will use small, real examples to visually demonstrate when the model should distinguish (entity vs. entity) and when it should group (synonyms, column permutations)
> 3. `Make the trade-off visually obvious.` We will redesign Figure 2 so that the tension between the two goals is explicit (e.g., emphasizing how forcing consistency too broadly harms row-level separation, and vice versa). The updated figure will highlight the specific structural and domain cases where each desideratum dominates.
>
> By restructuring the section and replacing dense, abstract diagrams with concrete examples and explicit trade-off illustrations, the revised paper will offer a much more accessible and intuitive explanation that aligns closely with the reviewer’s request.
>
>
> **C2 - Operational Clarification ($H_{\text{ctx}}$, $V_{\text{ctx}}$ extraction)**
>
> We thank the reviewer for pointing out that the current description of $H_{\text{ctx}}$ and $V_{\text{ctx}}$ is too implicit. Below we spell out the exact procedure with a concrete example; we will integrate this clarification into the revised paper.
>
> 1. `Tokenization.` Consider a row with a single header–value pair:
>     * Raw row: "datecreated: 2000 12 31"
>     * Tokenized input: $[\texttt{[CLS]}, \texttt{date}, \texttt{created}, \texttt{:}, \texttt{2000}, \texttt{12}, \texttt{31}, \texttt{[SEP]}]$
>     * Header-token indices: $P_h=\{\texttt{datecreated}: [1, 2]\}$
>     * Value-token indices: $P_v=\{\texttt{2000 12 31}: [4, 5, 6]\}$
>
>     The colon : is treated as a delimiter token and not included in the header span; similarly, $[\texttt{CLS}]$, $[\texttt{SEP}]$ are excluded from both spans. For a segment like "director: danny", the header span covers only the tokens of director (and its subword pieces, if any), not the colon.
> 2. `Embedding and contextualization.` The initial embedding of each token $x_j^{(k)}$ in segment $k$ is $z_j^{(k)}$. The sequence of the initial embeddings is then passed through the transformer encoder, producing contextualized token embeddings $\mathbf{e}_t$.
> 3. `Pooling to obtain H_ctx, V_ctx.` For each segment $(r,h)$ we then compute:
>
> $$H_{\text{ctx}}(r,h) = \frac{1}{|P_h|} \sum_{p \in P_h} \mathbf{e}_{t_p},$$
>
> $$V_{\text{ctx}}(r,h) = \frac{1}{|P_v|} \sum_{p \in P_v} \mathbf{e}_{t_p}.$$
> (i.e., mean-pooling the contextualized token embeddings over their respective header and value spans)
>
> In the revision, we will (i) include an explicit example like the one above to concretely define header/value spans, (ii) clarify that delimiter and special tokens are excluded from pooling, and (iii) update Figure 4 to clearly label which plot corresponds to BERT.
>
>
> **Q3 - Clarification of the Softmax Denominator Typo**
>
> We thank the reviewer for pointing out this inconsistency. You are correct that the denominator in the MSM loss should not use an undefined variable $\mathbf{e}_v$. The intended formulation follows the standard masked language modeling (MLM) objective, where softmax is computed over the logit vector derived from the contextualized token embedding $\mathbf{e}_t$. We will revise the equation in the paper to reflect this. Importantly, this would be a notation-only correction and does not affect the implementation, training procedure, or reported results.

---

> > ### Comment · Reviewer_ysm5 · 2025-11-26
> >
> > I thank the authors for their detailed response. I have carefully reviewed both the rebuttal and the revised manuscript. The inclusion of preliminary results is encouraging; however, it would be preferable for these findings to be formally incorporated into the revised paper with comprehensive descriptions, analysis, and reported variance. I will maintain my current positive evaluation.

---

> > > ### Author Response · Authors · 2025-12-04
> > >
> > > We sincerely appreciate your acknowledgement of our responses. We have thoroughly revised the manuscript to incorporate your suggestions. Please refer to our executive summary to the area chair for the point-by-point mapping of revisions.

---

### Official Review · Reviewer_rKbM · 2025-10-31

**Soundness:** 3
**Presentation:** 3
**Contribution:** 2
**Rating:** 2
**Confidence:** 4

**Summary:**

This paper proposes NAVI, a model that treats each *header–value segment* (header:value) as the atomic unit for table representation. NAVI integrates (a) a global header encoder, (b) Structure-aware Masked Segment Modeling (SMSM) for balanced masking of headers, values, and tokens, and (c) Entropy-driven Segment Alignment (ESA) for contrastive routing between low-entropy header-centric and high-entropy row-centric representations. The goal is to achieve fidelity (schema–value preservation and row distinctiveness) and consistency (robustness to schema, lexical, and structural variation). The paper provides theoretical analyses, extensive ablations, and evaluations on two WDC WebTables domains (Movie and Product).

**Strengths:**

- Well-motivated segment-level design. Treating (header:value) pairs as set elements with local positional encoding nicely combines permutation invariance and context awareness. The connection to DeepSets provides theoretical clarity.
- Entropy-based contrastive routing. The distinction between low- and high-entropy columns for header vs. row alignment is intuitive and empirically supported. The alignment/uniformity analysis is a good step toward geometric justification.
- Comprehensive evaluation axes. The paper evaluates discriminative (classification, clustering), generative (header prediction, value imputation), and invariance-based (PSI, header clustering) tasks, providing a holistic empirical picture.

**Weaknesses:**

- Limited domain diversity and possible bias.
   Experiments are restricted to two domains (Movies, Products) with the largest 100 tables per domain. This selection favors clean, high-quality schemas. The model’s robustness to smaller or noisier tables, numeric-heavy domains (e.g., finance), and unseen domains is unclear. Cross-domain and noisy-table experiments are needed.

- Under-specified training regimen.
   All models are trained for only 2 epochs with batch size 32 on datasets up to 3.9M rows. Such limited training may hinder convergence, confounding architectural effects with optimization noise. The paper should report learning curves, seed variance, and matched compute comparisons for baselines.

- Missing negative sampling details.
   InfoNCE-based contrastive results are sensitive to negative sample quality. Clarify negative sampling strategy (same/different table, same domain, batch size, memory bank use) and ablate negative set size.

- No comparison to classical tabular methods.
   The paper reports XGBoost only on top of embeddings. End-to-end baselines (e.g., XGBoost on raw features, TabPFN, TabNet) are missing, making it difficult to gauge absolute improvements.

**Questions:**

- Are header encoder parameters updated during training? Please include an ablation comparing frozen, partially fine-tuned, and lightweight alternatives.
- How sensitive is entropy-based routing to threshold settings? Compare fixed, percentile, and soft routing schemes.
- Could the author(s) ablate negative set size and temperature parameters (τ_dom, τ_ent) to analyze their effect on alignment/imputation?
- How does NAVI perform on numeric-heavy domains? Compare against numeric-specialized models (e.g., TP-BERTa, TabPFN).
- Can author(s) demonstrate cross-domain transfer (e.g. Product→Movie, Movie→Product) to substantiate domain-invariant anchor learning?
- How robust is NAVI under schema noise (renaming, typos, column swaps)?


I would consider raising my score if the authors can adequately address these questions.

---

> ### Author Response · Authors · 2025-11-21
>
> We thank Reviewer rKbM for their thorough and constructive feedback. We appreciate their recognition of our "*well-motivated segment-level design*", the "*intuitive entropy-based contrastive routing*" mechanism, and our "*comprehensive evaluation*" across discriminative, generative, and invariance-based tasks. These strengths validate our core contributions and motivate us to carefully address the reviewer's remaining concerns.
>
> In our initial responses, we provide point-by-point executive summaries of our answers and clarifications:
>
> **Executive Summaries:**
>
> 1. **Training Regimen & Convergence**: We clarify the rationale behind the two-epoch training schedule, and provide matched-compute comparisons to ensure architectural fairness.
>
> 2. **Sampling & Routing in ESA (Entropy-driven Segment Alignment)**: We detail negative sampling sources, justify our quartile-based routing choice, and provide new ablations on negative size, temperature asymmetry, and entropy thresholds.
>
> 3. **Scope Clarification & Extension with Numeric Data**: We clarify NAVI’s intended domain (symbolic, text-heavy web tables), compare against numeric-specialized baselines, and discuss principled extensions for numeric attributes.
>
> 4. **Robustness to Domain and Schema**: We present new results for zero-shot cross-domain transfer and schema-noise robustness (synonyms, typos, and column reordering), demonstrating NAVI’s stability under real-world perturbations.

---

> ### Author Response · Authors · 2025-11-21
>
> **W2, Q1 - Clarification of Training Regimen**
>
> `Justification of 2 Epochs (W2).` We observe that NAVI achieves most of its performance gains by epoch 2. While the improvements from epoch 2 to epoch 3 vary across domains, further training does not consistently improve downstream accuracy, and in several cases (e.g., Product Cls), performance begins to degrade, suggesting early signs of overfitting. To address the reviewer’s concern, we will include full learning-curve plots in the revised paper.
>
> `Stability of Global Header Encoder Updates (Q1).` The global header encoder must generalize across heterogeneous tables while adapting to domain-specific usage. Freezing it tests pure transfer, whereas partial and full fine-tuning allow controlled adaptation of low-entropy header semantics. Our ablations show a consistent ordering frozen < partial < full across tasks (Table 1), confirming that full training provides the best domain coherence and downstream accuracy. These results justify our choice to update the header encoder in the main experiments.
>
> (Table 1)
> |  | Prod (Cls) | Prod (Val loss) | Mov (Cls) | Mov (Val loss) |
> |-------|------------------|-------------------|------------------|--------------------|
> | Default (E2, Header Enc Fully Trained) |   **0.8261**   |    **1.1691**  |  **0.3679**   |  1.4227    |
> | E1     |  0.8098    |   1.8574   |  0.2477   |   1.6257   |
> | E3     |  0.8202    |   1.3951   |  0.2867   |  **1.2366**    |
> | Header Encoder Frozen     |  0.7864    |   3.1365   |   0.3258  |   2.4171   |
> | Header Encoder Partially Trained     |  0.7902    |   1.4400   |   0.2910  |   1.4378   |
>
> `Matched Compute Comparisons (W2).` To ensure that architectural differences—not optimization budget—drive performance gaps, we trained all baselines under strictly matched compute. Specifically, every model (BERT, HAETAE, TAPAS, NAVI) was trained for exactly 10,276 update steps, using batch size 32 with 2× gradient accumulation, identical learning-rate schedules, and identical optimizer configurations. This guarantees that all methods receive the same number of parameter updates, regardless of model size or training throughput.

---

> ### Author Response · Authors · 2025-11-21
>
> **W3, Q2, Q3 - Details of Sampling, Routing and Ablating Its Impact**
>
> `Negative Sample Sources (W3).` Our negative sampling scheme is aligned with the semantics of ESA: low-entropy contrast draws in-batch header negatives from the same table to refine universal header embeddings, whereas high-entropy contrast uses row-level negatives from the same table to strengthen instance discrimination. This design eliminates the need for memory banks while ensuring sufficient negative diversity. To make this concrete, each batch contains 4 tables, and we sample 16 rows per table (effective negative set size = 16). For a given segment:
> * *Low-entropy segments* use as negatives the other headers from its own table
> * *High-entropy segments* use as negatives the values from other rows within the same table.
>
> Batching by tables in this way provides balanced learning while preserving table-level semantics for both contrastive objectives.
>
> `Routing Strategy (Q2).` We adopt a quartile-based routing as it is the simplest strategy reflecting our core motivation for consistent alignment. In real-world web tables, low-entropy columns contain frequently repeated, domain-coherent categories (e.g., genre, country), while high-entropy columns contain near-unique identifiers (e.g., title, url). Mid-entropy columns, by contrast, often combine long-tailed, sparsely repeated, and distributionally noisy values, making their empirical entropy unstable and unsuitable for consistent alignment. Routing these ambiguous columns risks pushing them toward incorrect semantic anchors. We acknowledge that routing with various entropy thresholds or advanced soft weighting function can be potential extensions, requiring extra fine-tuning upon datasets to boost performance. We believe a simple quartile-based routhing practically preserves interpretability and analytic clarity, demonstrating NAVI’s core design. Please refer to the relevant analysis in the next response.
>
> `Ablations on Negative Size, Temperature, and Entropy Threshold (Q2, Q3).` Our negative-set ablation shows that while larger batches naturally introduce more negatives, performance does not improve monotonically. Instead, an effective negative set size of 16 offers the most balanced and consistent results across domains, suggesting that the optimal negative diversity is domain-dependent but moderate batch sizes yield the most stable behavior.
> More importantly, our new entropy-threshold experiments validate the core assumption behind ESA. Although our default choice (Q1/Q3) follows a standard and widely used percentile split, we find that pushing the thresholds further toward the extremes (e.g., 10p/90p)—thereby enlarging the mid-entropy “dead zone” and restricting alignment to the most stable regions—often improves performance (Table 2). This aligns with our motivation that only extreme entropy regimes provide reliable semantic signals for alignment, while mid-entropy columns are better left to MSM rather than contrastive learning. Taken together, these results show that ESA’s effectiveness stems from its semantic treatment of entropy extremes rather than from any specific threshold choice. This also leaves room for dataset-specific fine-tuning of routing boundaries. We will add these findings to the revised manuscript.
> The temperature ablations further support the semantic asymmetry built into ESA. Low-entropy (domain-coherent) columns require a softer temperature (0.13) to keep similar categories close, while high-entropy (entity-discriminative) columns require a sharper one (0.07) to maintain row-level separability. As shown in Table 2, enforcing symmetric temperatures—either both sharp (0.07/0.07) or both soft (0.13/0.13)—produces unstable performance patterns across domains. This variability indicates that no symmetric setting works reliably across domains. In contrast, the asymmetric temperature configuration (0.07 / 0.13) provides the most domain-robust and consistent behavior, reinforcing ESA’s semantic motivation and justifying the default choice used in our main experiments.
>
> (Table 2)
> | Ablations                      | Prod (Cls) | Prod (Imp) | Mov (Cls) | Mov (Imp) |
> |--------------------|----------|----------|---------|---------|
> | Default (Q1/Q2, 0.07/0.13, 16)|     0.8261 |       0.7295   |    0.3679     |  0.6870  |
> | Neg size 8           |  **0.8441** | **0.7695**   |    0.2454     |  0.6866  |
> | Neg size 32          |     0.8118 |  0.7472   |    0.3588     |  0.6251  |
> |Entropy threshold 10p/90p     |   0.8268 |    0.7610   |   0.3798     |  **0.6889**  |
> | Entropy threshold 40p/60p     |     0.8056 |       0.7192   |    0.2236     |  0.6760  |
> | Entropy threshold 50p         |     0.7805 |       0.7118   |    0.1852     |  0.6482  |
> | Temp. ent/dom - 0.07/0.07     |     0.6317 |       0.7469   |    **0.4343**     |  0.6779  |
> | Temp. ent/dom - 0.13/0.13     |     0.6353 |       0.7176   |    0.2357     |  0.6758  |

---

> ### Author Response · Authors · 2025-11-21
>
> **W4, Q4 - Scope Clarification and Enhancement of Comparison**
>
> `Scope Clarification.` NAVI is intentionally designed for symbolic, text-heavy web tables with domain-specific attributes (e.g., WDC web tables), consisted of rich header–value semantics reflected in a nominal form. Classical machine learning methodologies (e.g., GBDTs) or numeric-specialized models (e.g., TP-BERTa with magnitude-aware tokenization) handle numerical attributes effectively, but their generalization across non-numerical columns, schema variations, and heterogeneous tables within a single domain remains an open problem. Language models excel at capturing complex symbolic semantics, yet they fundamentally struggle to perceive bi-directional table structure, permutation invariance, and relationships among tables in a database. NAVI’s contribution lies precisely in leveraging the semantic strengths of language models while addressing these structural limitations through segment-level modeling and entropy-driven alignment. Numeric-heavy datasets require additional type-aware coding (e.g., discretization, magnitude encodings), which is outside of our current scope. We believe that there are a lot of exciting ways to extend NAVI to deal with numerical attributes, which is further discussed in the following response.
>
>
> `Comparison with Classical and Hybrid Baselines (W4, Q4).` To contextualize NAVI’s contribution beyond LM-based table encoders, we additionally compare against classical end-to-end tabular learners trained directly on raw features, including XGBoost and TabPFN (Table 3). These baselines establish the performance ceiling expected for clean, numeric-aware pipelines. As shown, raw-feature XGBoost and TabPFN perform strongly on representative tabular task, classification; however, NAVI’s goal is not to replace such models but to address the long-standing challenges faced by symbolic, text-heavy web tables—domains where numeric-specialized pipelines offer limited leverage.
> To illustrate NAVI’s interoperability with existing numerical pipelines, we include a minimal hybrid variant, NAVI_concat, which simply appends raw numerical columns to NAVI’s representations. Despite its naïveté, this prototype already exceeds XGBoost with raw features in the Product domain, suggesting that more principled type-aware fusion could be highly effective. Finally, TableVectorizer, a robust feature-engineering pipeline that mixes scalable numeric preprocessing with text embeddings produced from SentenceTransformer, achieves the strongest results—highlighting that NAVI can serve as a drop-in replacement for the LM component in such pipelines. These comparisons clarify that NAVI is complementary to classical numeric-aware models, while providing the structural and semantic robustness necessary for symbolic, schema-diverse web tables, which remain the central focus of this work.
>
> (Table 3)
> |                       | Prod (XGB) | Prod (TabPFN) | Mov (XGB) | Mov (TabPFN) |
> |-------------------------------|----------|----------|---------|---------|
> | Only-numeric (Raw feature) |     0.8880 |       0.8825   |    0.4644     |  0.4990  |
> Only text embedding (NAVI)|     0.8261 |       0.8645   |    0.3679     |  0.4177  |
> Text embedding + numeric (NAVI+Numeric Raw Feature)|     0.8932 |       0.8601   |    0.4041     |  0.4057  |
> Feature engineering for feature types: numeric, datetime, low-cardinality text, high-cardinality text (TableVectorizer)|     0.9306 |       0.9376   |    0.6263     |  X (dim>2000)  |

---

> ### Author Response · Authors · 2025-11-21
>
> **W1, Q5, Q6 - Robustness Across Domains and Schema Variations**
>
> `Cross-Domain Transfer.` To assess whether NAVI learns domain-invariant header anchors, we conduct zero-shot transfer between Movie and Product tables (Table 4). Movie → Product achieves 0.838 classification accuracy (slightly higher than Product in-domain 0.826) and retains ~66% imputation performance. Product → Movie preserves ~78% (classification) and ~74% (imputation) of Movie in-domain accuracy. These results show that low-entropy headers (e.g., rating, category) form stable semantic anchors that generalize across domains, even without cross-domain pretraining. ESA’s separation of domain-coherent (low-entropy) and entity-specific (high-entropy) segments enables this robustness, providing empirical evidence of domain-invariant anchor learning.
>
> (Table 4)
> | M->P (Cls) | P->P (Cls)  | M->P (Imp) | P->P (Imp) | P->M (Cls) | M->M (Cls) | P->M (Imp) | M->M (Imp) |
> |----------|----------|---------|---------|----------|----------|---------|---------|
> | **0.838** | 0.826 | 0.481 | **0.730** | 0.329 | **0.422** | 0.508 | **0.687** |
>
>
> `Schema Noise Robustness.` Our Consistency Analysis (Section 3.3) shows that NAVI produces highly stable embeddings under schema noise—including lexical variation (synonyms) and structural diversity (column-order changes)—demonstrating near-zero permutation sensitivity and strong lexical invariance. To further quantify robustness, we evaluate NAVI in the Fidelity Analysis setting using lexically and structurally perturbed table variants. . We applied three types of perturbations:
> 1. *Synonym replacement*: For each table, we identified low-entropy headers then randomly sampled 50% of these low-entropy headers and replaced them with semantically equivalent synonyms from a curated mapping (e.g., "director.name" → "auteur.name"). Synonym selection was random from the available alternatives for each header.
> 2. *Header typos*: We randomly sampled 50% of low-entropy headers (identified as above) and introduced character-level corruption by randomly applying 1-2 operations per header: character substitution, insertion, or deletion. Within each table, the same header was consistently corrupted to the same variant to maintain structural coherence.
> 3. *Column-permuted rows*: For each row independently, we randomly shuffled the column order while preserving header-value pairs, simulating arbitrary structural reordering without semantic changes.
>
> NAVI was evaluated under all three perturbation types on both classification (F1 score with XGBoost) and imputation (value prediction accuracy) tasks for Product and Movie domains, using models trained on clean schemas. Under synonym replacement and column reordering, NAVI consistently maintains strong performance—often matching or slightly exceeding the clean-schema baseline. This stability aligns with our Consistency Analysis: lexical variants are absorbed by the global header encoder, and structural permutations have minimal effect because NAVI encodes rows as unordered sets of header–value segments. As expected, header typos introduce the largest degradation, since character-level corruption alters subword tokens and weakens header–value grounding; nevertheless, NAVI still preserves a substantial portion of its clean-schema performance (e.g., Product Cls: 0.8261 → 0.7899; Movie Imp: 0.6870 → 0.6384). Overall, these results confirm that NAVI is robust to common schema inconsistencies—particularly renamings and column reordering—while degrading gracefully under more severe corruption such as typos.
>
> (Table 5)
> | | Prod (Cls) | Prod (Imp) | Mov (Cls) | Mov (Imp) |
> |----------|----------|---------|---------|---------|
> |  Default  |     0.8261     | 	0.7295  |  0.3679  | 0.6870
> |  Synonym  |  0.8262   |   0.7193   |  0.3758  | 0.6734
> |   Typo       |   0.7899    |  0.6541   |  0.3691   | 0.6384
> |   Column Reordered      |   0.8311       |    0.7373     |      0.3961   | 0.6817

---

### Official Review · Reviewer_3n4C · 2025-11-01

**Soundness:** 3
**Presentation:** 2
**Contribution:** 2
**Rating:** 4
**Confidence:** 4

**Summary:**

This paper proposes NAVI: Entropy-aware Alignment via Header–Value Induction. NAVI captures the structural properties of tables through schema-aware segment induction and modeling. In addition, NAVI employs entropy-driven alignment of segments to selectively incorporate domain knowledge shared among in-domain tables. Through various experiments, the paper shows effectiveness of NAVI on various downstream tasks.

**Strengths:**

In general, the paper is easy to follow and the paper provides theoretical grounds on constructing the proposed method.

**Weaknesses:**

-	The figures does not help understand the proposed method.
-	In figure 1, the readers cannot see what they are supposed understand. Also, clear explanation with concrete examples for distinctiveness (fidelity) and robustness (consistency) is required.
-	In figure 2, the paper states there are trade-offs, but it is really hard to visualize what the trade-offs are.
-	It would be great to have explanations of the concepts with the examples shown in the figures.
-	As the paper addresses the importance of table representation for downstream tasks, it would be interesting to see how NAVI compares to simple heuristics for encoding tables (eg., Tablevectorizer or TextEncoder in Skrub package) combined with tabular learning methods such as TabPFN, XGB, and LR.

**Questions:**

-	What characterizes the distinctiveness (fidelity) and robustness (consistency)? What are some concrete examples?
-	How does NAVI deal with numerical values?
-	Would there be more datasets to compare the performance of NAVI?
-	What is the ground for choosing Bert-style model? Could NAVI benefit from a more sophisticated architecture?

---

> ### Author Response · Authors · 2025-11-21
>
> We thank Reviewer 3n4C for their constructive feedback and their positive assessment of the paper’s soundness and conceptual contribution. We appreciate the specific suggestions regarding clarity of figures and the request for additional baselines and examples. Below, we address each concern point-by-point.
>
> **W1, W2, W3, W4, Q1 - Clarifying Fidelity and Consistency in Table Representation**
>
> A key motivation for formalizing fidelity and consistency is that, while numerical attributes are effectively handled by classical ML pipelines (e.g., GBDTs) or numeric-specialized LM variants (e.g., TP-BERTa), non-numerical columns and symbolic schema variation remain far more difficult for language models to generalize over. LMs excel at abstract semantic reasoning but inherently struggle to capture bi-directional table structure, permutation invariance, and cross-table relational consistency—blind spots that become critical in multi-table in-domain corpora such as WDC. NAVI is designed precisely to bridge this gap: it leverages the semantic strengths of LMs while correcting their structural weaknesses through segment-level induction and entropy-aware alignment. **At the highest level, fidelity and consistency simply answer the fundamental question of table representation: what should differ and what should remain the same across in-domain tables.** This motivates the need for clear desiderata that characterize what table representations must preserve and what they should be invariant to. This motivates the need for clear desiderata that characterize what table representations should preserve and what they should be invariant to.
>
> To address the reviewer’s request for concrete and intuitive definitions, we clarify fidelity and consistency along two complementary axes—structural and domain-level—both of which are challenging for language models to capture when operating on tabular data.
>
> *Fidelity: Preserving Fine-Grained, Discriminative Semantics.* Fidelity concerns what the model must remain sensitive to—the distinctions that matter for identifying structural roles or differentiating entities.
> * *Structural Fidelity*: Linearizing a table into a sequence often obscures structural cues. Structural fidelity requires *preserving each element’s functional role* despite linearization—for example, recognizing in “genre, director, action, tim” that genre/director are headers and action/tim their values, rather than treating them as undifferentiated tokens.
> * *Domain Fidelity*: Within domains like movies or products, many rows share similar schema-level attributes. Domain fidelity ensures *entity-level discriminability*, keeping rows distinct even when they share features (e.g., two action films with overlapping casts should not collapse in embedding space).
>
> *Consistency: Being Robust to Superficial Variation While Preserving Shared Concepts.* Consistency concerns what the model should be invariant to—variations that should not alter underlying meaning.
> * *Structural Consistency*: Language models are overly sensitive to formatting variations such as column order, row order, or column count. Structural consistency requires invariance to these *superficial rearrangements*, ensuring that moving “genre” from column 2 to column 7 does not alter its learned semantics.
> * *Domain Consistency*: Domain consistency requires a *stable understanding of domain concepts* across heterogeneous tables, even when schema or naming differ. This includes treating lexical variants like “country,” “nation,” and “production_country” as the same concept and mapping genre values (“action,” “drama,” “fantasy”) to a coherent semantic space. This stability enables robust generalization across diverse tables.
>
> To directly address the reviewer’s concern regarding clarity, we will revise Figures 1 and 2 to visually reflect these definitions.

---

> ### Author Response · Authors · 2025-11-21
>
> **Q2 - Handling of Numerical Values**
>
> NAVI currently treats numerical values as symbolic token sequences, meaning it does not encode ordinal or metric relationships (e.g., 10 ≈ 11 ≪ 100). This is a design choice aligned with our focus on text-heavy, symbolic web tables where semantic structure and schema variation dominate the learning challenge. As a result, NAVI does not perform dedicated numeric reasoning, and numeric-heavy domains are better addressed with models specifically designed for magnitude-aware processing.
>
> We provide a detailed technical discussion of this limitation and possible extensions—such as integrating numeric-specialized classical approach (e.g., TabPFN) and our method into a single feature engineering pipeline or adding type-aware numeric modules—in our rebuttal to Reviewer 4, where this issue is the primary point of evaluation. We will clarify this scope and future direction in the revised manuscript.
>
> **W5 - Comparison With Simple Heuristics**
>
> To address the reviewer’s request for heuristic encoders, we evaluated Skrub’s TableVectorizer and TextEncoder alongside NAVI. TextEncoder operates only on individual columns and cannot serve as a unified row encoder. TableVectorizer—while applicable—functions as a feature-engineering pipeline that mixes scalers, one-hot encoders, hashing vectorizers, and text embeddings, producing supervised feature matrices rather than unified row representations.
>
> Nonetheless, we include TableVectorizer in our comparisons for completeness (Table 1).
> We highlight three findings:
> 1. Heuristic pipelines can be strong. As shown in the table, TableVectorizer achieves the highest raw performance, even when paired with a general SentenceTransformer.
> 2. But they produce >2000-dimensional vectors in product dataset, making them incompatible with compact models such as TabPFN without further reduction. NAVI, by contrast, produces compact 768-dimensional embeddings based on a BERT backbone (and could benefit even further from stronger encoders such as ModernBERT).
> 3. NAVI benefits from incorporating numerical signals: A simple hybrid (NAVI_concat) that naively concatenates raw numeric columns to NAVI embeddings already surpasses XGBoost on raw features in the Product domain, suggesting that principled integration (e.g., replacing SentenceTransformer inside TableVectorizer with NAVI) is an immediately promising next step.
>
> (Table 1)
> |                       | Prod (XGB) | Prod (TabPFN) | Mov (XGB) | Mov (TabPFN) |
> |-------------------------------|----------|----------|---------|---------|
> | Only-numeric (Raw feature) |     0.8880 |       0.8825   |    0.4644     |  0.4990  |
> Only text embedding (NAVI)|     0.8261 |       0.8645   |    0.3679     |  0.4177  |
> Text embedding + numeric (NAVI+Numeric Raw Feature)|     0.8932 |       0.8601   |    0.4041     |  0.4057  |
> Feature engineering for feature types: numeric, datetime, low-cardinality text, high-cardinality text (TableVectorizer)|     0.9306 |       0.9376   |    0.6263     |  X (dim>2000)  |
>
> **Q4 - Choice of BERT-Style Architecture**
>
> We use a BERT-style encoder because header–value segments resemble short textual sequences, where BERT performs strongly and the architecture provides a stable backbone for isolating the effects of SSI (Schema-aware Segment Induction), MSM (Masked Segment Modeling), and ESA (Entropy-driven Segment Alignment).
>
> We agree that more sophisticated architectures (e.g., long-context transformers like Modern BERT) could further improve performance. Our goal in this work was to demonstrate that semantic induction + entropy-based routing itself provides value, independent of backbone complexity. We plan to explore architectural variations in future work.

---

### Author Response · Authors · 2025-12-04
**Executive summary of the rebuttal and revision**

Dear Area Chair,

We would like to thank you for your time and efforts in reviewing our submission. We are also deeply grateful to the reviewers for their constructive feedback and suggestions, which have helped further enhance our work. In the revision of our manuscript, we have fully incorporated the reviewers' comments and clarification made during the rebuttal period. As two reviewers who responded to our response before the unexpected event acknowledged, we successfully clarified the problem setup, refined the methodological exposition, and strengthened the empirical validation.

In brief, our clarifications and corresponding revisions fall into three major categories:
1. Clarification of core concepts and methodological details
2. Corrections to configurations and updated experimental results
3. Additional analyses and supplementary experiments

A point-by-point summary of the specific actions taken and the corresponding sections of the revised manuscript is provided in the table below. We are particularly glad that these revisions make the contribution of our work clearer and our evaluation results more robust, owing to the constructive feedback and questions from the reviewers. Given this unexpected circumstance, we sincerely appreciate your careful consideration in reassessing the paper in light of our detailed responses to reviewers and the updated manuscript.

Sincerely,

The authors



|Actions Taken for Reviewer's Comments|Revised Parts in Manuscript|
|-|-|
|***Conceptual and Methodological Clarification***|
|Clarification of definition of fidelity and consistency with refined figure and concrete examples|Section 1|
|Clarification of the theoretical domain properites|Appendix A.2|
|Clarification of our scope as symbolic understanding of heterogeneous in-domain tables|Section 1|
|Updated figure for the overall method|Section 2|
|Refinement of ambiguous notations and expanded explanations of contextual pooling|Section 2.1|
|Fixed typo in objective notation for Masked Segment Modeling|Section 2.2|
|***Configuration Corrections***|
|Correction of entropy thresholds, contrastive temperatures, and alignment weights|Section 2.3, 3.1|
|Updated results across Tables 1-4 after applying corrections|Section 3.2, 3.3, 3.4|
|***Supplementary Experiments***|
|Baseline extensions for classification task|Section 3.2|
|Evaluation of fidelity of NAVI under schema noise|Appendix C.1|
|Justification of training regimen and hyperparameters|Appendix D.4|
|Refinement of geometric analysis|Section 3.5, Appendix C.2|

---

### Meta-Review · Area_Chair_cGWe · 2026-01-06

**Summary:**

This submission generated much discussion. The reviewers had some comments about the clarity of the manuscript and the figures, which seemed to lead to improvements.
The method is based on sophisticated considerations, but the reviewers required simple and supervised baselines, and simple heuristics "TableVectorizer" performed markedly better than the contributed method.

Also, the reviewers raised the point that the method considers numbers as tokens, thus discarding their topology, which is probably a sizeable limitation given their importance in tables.

**Reviewer Concerns:**

There were many back and forth, and some points were addressed. However, the reviewers' concerns on simple baselines led to more questioning, given that these performed really well.

**Reviewer Scores:**

Given the results of the complementary analysis and what I read in the final remarks of the reviewers, I do not believe that the reviewers would have strongly increased their ratings.

---

### Decision · Program_Chairs · 2026-01-26

Reject